# LatentEvolve: Self-Evolving Test-Time Scaling in Latent Space

## Abstract

Test-time Scaling (TTS) has been demonstrated to significantly enhance the reasoning capabilities of Large Language Models (LLMs) during the inference phase without altering model parameters. However, existing TTS methods are largely independent, implying that LLMs have not yet evolved to progressively learn how to scale more effectively. With the objective of evolving LLMs to learn "how to scale test-time computation," we propose `LatentEvolve`, a self-evolving latent TTS framework inspired by the complementary learning system (CLS) theory. Analogous to the human brain's dual system of a fast-recall hippocampus and a slow-consolidating neocortex, `LatentEvolve` comprises two evolutionary components: *daytime scaling*, which rapidly retrieves historical latent representations to better guide current LLM reasoning; and *nighttime scaling*, which integrates past latent optimizations in a manner akin to the human brain's consolidation of experiences during sleep. The alternation of daytime and nighttime processes facilitates a fast and slow evolution of LLM TTS, mirroring human cognitive dynamics in a fully unsupervised manner. Extensive experiments across eight benchmarks and five model backbones demonstrate that our `LatentEvolve` surpasses state-of-the-art TTS methods such as LatentSeek and TTRL by up to $13.33\%$ and exhibits exceptional cross-domain and cross-backbone generalization. The codes are available at https://anonymous.4open.science/r/latent-evolve/.

## 1 Introduction

The general capabilities of large language models (LLMs) have been extensively developed and widely recognized across numerous domains, such as mathematical reasoning (Zeng et al., 2024; Wu et al., 2025), software engineering (Wei et al., 2025; Luo et al., 2025; Yang et al., 2024), multimodal understanding (Zheng et al., 2025b; Su et al., 2025), and embodied action (Wang et al., 2023a), emerging as dominant paradigms that are steadily advancing toward artificial general intelligence (AGI) (Bubeck et al., 2023). Much of this success in recent years has been driven by *training-time scaling*, wherein increasing the volume of training data and parameters consistently yields performance improvements (Kaplan et al., 2020; Aghajanyan et al., 2023). However, the pace of this scaling, particularly in terms of pre-training scale, has begun to slow, constrained by its resource-intensive nature and the depletion of high-quality training data (Villalobos et al., 2022; Zhou et al., 2025). Consequently, a growing body of research has shifted attention to *test-time scaling* (TTS) (Zhang et al., 2025c; Chung et al., 2025), aiming to fully harness the intrinsic knowledge of LLMs to maximize their real-world utility without additional training during the test phase.

The dimensions of TTS are highly diverse. One prominent form is **(I) parallel scaling**, wherein multiple candidate responses are generated for a given query, which are subsequently aggregated via an appropriate mechanism. This can involve multiple samples from a single LLM (Brown et al., 2024; Snell et al., 2024) or sampling from multiple heterogeneous LLMs (Zhang et al., 2025d; Ye et al., 2025). Another form is **(II) sequential scaling**, where the LLM iteratively refines solutions based on its own previous outputs, and which underlies many "System 2"-style generation methods (Yu et al., 2024; Wei et al., 2023; He et al., 2024; Gou et al., 2024). Other variants include *hybrid* approaches that integrate both strategies (Wang et al., 2024a; Besta et al., 2024), as well as *internalized scaling*, where models like DeepSeek R1 (Guo et al., 2025) and OpenAI o-series (Li et al., 2025b) are inherently capable of adaptively allocating computational resources during inference.

Figure 1: The daytime scaling of `LatentEvolve` functions analogously to the human hippocampus, rapidly retrieving memory cues, whereas the nighttime scaling mirrors the neocortex during sleep, performing deep integration of accumulated knowledge.

However, regardless of the specific form, most TTS paradigms lack the capacity for *self-evolution*, as inference-time computations for distinct queries are typically treated as mutually **independent** events. For example, in verbal reinforcement learning approaches such as Reflexion (Shinn et al., 2023) and Mind Evolution (Lee et al., 2025), successful reflective strategies are instance-specific and are not transferred to subsequent tasks. Likewise, in "sampling-and-voting" scaling methods (Brown et al., 2024; Irvine et al., 2023), prior successes in selecting the correct answer do not inform or refine future selection strategies. This *inter-task independence* fundamentally constrains the potential of TTS paradigms to progressively evolve through continual interaction with the environment. This raises a natural yet critical research question: *How can we design a TTS framework that learns from experience, enabling its scaling capabilities to evolve and improve as it solves more problems?*

To address this challenge, we introduce `LatentEvolve`, a self-evolving TTS framework inspired by the Complementary Learning Systems (CLS) theory (McClelland et al., 1995; Kumaran et al., 2016). CLS theory posits that the brain uses two synergistic systems: a fast-learning hippocampus for specific episodic memories, and a slow-learning neocortex for consolidating these experiences into general knowledge. Analogously, `LatentEvolve` operates through a dual-phase evolution:

- ☀ **Daytime Scaling** for *fast, episodic adaptation*: For each new query, `LatentEvolve` performs instance-level latent optimization that steers the LLM toward better reasoning paths. This process is initialized by retrieving relevant "episodic traces", *i.e.*, latent representations from previously solved problems, mirroring the *daytime* fast recall of individual memories.

- ☁ **Nighttime Scaling** for *slow, procedural consolidation*: Mirroring how the brain consolidates experiences into general skills during sleep, `LatentEvolve` periodically fine-tunes a compact knowledge consolidation model (*latent weaver*) on the collection of *daytime* traces. This *nighttime* process distills these specific experiences into procedural knowledge, evolving to generate superior initial latent representations for future tasks.

Within this continual interplay, `LatentEvolve` enables LLMs to perform test-time computation during daytime inference while simultaneously accumulating experiential knowledge. During nighttime reflection, these experiences are periodically consolidated into endogenous procedural memory, thereby achieving a "fast-slow" evolution of test-time scaling. The entire process operates **without** reliance on ground-truth labels or any other external signals.

**Experimental Observation.** Extensive evaluations across eight benchmarks spanning four domains demonstrate that `LatentEvolve` provides: ❶ **high performance**: achieving up to 23.3% gains on math reasoning, surpassing GRPO and LatentSeek on MATH-500 by 1.75% and 11.40%, respectively; ❷ **cross-domain generalization**: test-time scaling on MMLU and MATH transfers to out-of-domain datasets, yielding gains of 7.07% on GPQA and 5.22% on JAMA; ❸ **continual learning ability**: test-time scaling across multiple new domains does not degrade performance on previously seen domains and can even provide modest improvements.

## 2 RELATED WORK

**Test-time computation** is a canonical pathway for transitioning from System 1 to System 2 models, with two primary branches: *test-time training* (TTT) and *test-time scaling* (TTS). The former involves updating model parameters during the test phase in an unsupervised manner, as exemplified

by TTT (Sun et al., 2020; Akyürek et al., 2024), TTT+++ (Liu et al., 2021), and SIFT (Hübotter et al., 2025). The latter increases computational expenditure without altering parameters, which can occur in the **(I) explicit, natural-language space**, as in self-correction (Shinn et al., 2023; Gou et al., 2024; Kang et al., 2025), feedback modeling (Cobbe et al., 2021; Yu et al., 2025), or repeated sampling (Gui et al., 2024; Ye et al., 2025); or it may operate in the **(II) latent space**, where methods such as Coconut (Hao et al., 2024) and SoftCoT (Xu et al., 2025b;c) perform deep scaling within the model's hidden representations. Our proposed `LatentEvolve` falls primarily within the latent TTS. Yet, regardless of form, existing approaches are rarely capable of rapid evolution through the ongoing process of problem solving, a limitation that `LatentEvolve` is designed to overcome.

**Latent Computation & Reasoning** seeks to exploit continuous latent representations, rather than discrete language space, to enable a more machine-native and concise form of reasoning for LLMs (Zhu et al., 2025). Mainstream approaches can be broadly categorized as: **(I) architecturally enabling native latent reasoning**, as exemplified by Coconut (Hao et al., 2024), CoLaR (Tan et al., 2025), and Recurrent Depth (Geiping et al., 2025); and **(II) employing latent computation to steer LLM generation**, as in LatentSeek (Li et al., 2025a), SoftCoT (Xu et al., 2025c;b), and otherss (Liu et al., 2024; Sun et al., 2025), which leverage latent representations as an intervention to modulate the quality of generated outputs. Other methods, such as IMM (Orlicki, 2025) and MemoryLLM (Wang et al., 2024c; 2025a), employ latent tokens as a means of preserving contextual memory. Distinct from these approaches, `LatentEvolve` implements a dual-stage test-time evolution within the latent space, whereas prior strategies generally remain inter-task independent.

**Self-Evolving LLM & Agent.** How to evolve LLMs during their interactions with the environment has drawn increasing attention from the research community (ang Gao et al., 2025; Fang et al., 2025). Existing approaches generally employ certain carriers for evolution, including: **(I) parametric update**, wherein prior experiences are encoded directly into model parameters (Zeng et al., 2023; Chen et al., 2024; Zhao et al., 2025; Chen et al., 2025b); **(II) experience databases**, in which past problem-solving trajectories (Zhao et al., 2024; Song et al., 2024) or distilled experiential knowledge (Zhang et al., 2025a; Wang et al., 2025b; Tang et al., 2025) are leveraged to contextually enhance LLM capabilities; and **(III) skill condensation**, where reusable tools (*e.g.*, APIs, MCPs) are encapsulated as functional assets (Zheng et al., 2025a; Suzgun et al., 2025; Zhang et al., 2025b; Qiu et al., 2025b;a). Distinct from these paradigms, `LatentEvolve` performs test-time evolution within the latent space, treating the latent sequences as a compact and adaptable skill repository.

## 3 PRELIMINARY

In this section, we formally describe the procedure of current latent-based TTS methods, which manage to steer LLM's generative process by introducing adaptable, continuous vectors.

**Latent-Space Aided Reasoning.** Let $\pi_{\boldsymbol{\theta}}$ be a language model with frozen parameters $\boldsymbol{\theta}$. For a given problem context $\mathbf{c}$, the standard generative process produces an output sequence $\mathbf{y}$ by sampling from the conditional probability distribution $p(\mathbf{y}|\mathbf{c};\boldsymbol{\theta})$. The core principle of this paradigm is to introduce an auxiliary sequence of continuous vectors, $\mathbf{z} = (z_1, z_2, \cdots, z_L)$, which we refer to as a *latent token sequence*. These vectors act as a dynamic, instance-specific control signal that conditions the generative process of the frozen LLM. The generation is thus reformulated as sampling from a new distribution, conditioned on both the original context and the latent intervention:

$$\mathbf{y} \sim p(\mathbf{y}|\mathbf{c}, \mathbf{z}; \boldsymbol{\theta}) \tag{1}$$

The latent sequence $\mathbf{z}$ can be introduced through various mechanisms, such as being prepended to input embeddings, directly augmenting the model's internal key-value (KV) cache, or representing a latent thought process for subsequent decoding (Xu et al., 2025c; Liu et al., 2024; Sun et al., 2025). The primary objective is to find an optimal latent intervention $\mathbf{z}^*$ that maximizes an objective function $J(\mathbf{z})$. This objective is formalized as the expected quality of the generated output:

$$\mathbf{z}^* = \arg\max_{\mathbf{z}} J(\mathbf{z}), \quad \text{where} \quad J(\mathbf{z}) = \mathbb{E}_{\mathbf{y} \sim p(\mathbf{y}|\mathbf{c}, \mathbf{z}; \boldsymbol{\theta})}[Q(\mathbf{y})] \tag{2}$$

where $Q(\mathbf{y})$ is a scoring function that evaluates the quality of an output $\mathbf{y}$.

**Generation of Latent Representations.** The mechanism for generating the latent sequence $\mathbf{z}$ defines the specific TTS method. Existing work either optimizes a single set of *task-specific* soft

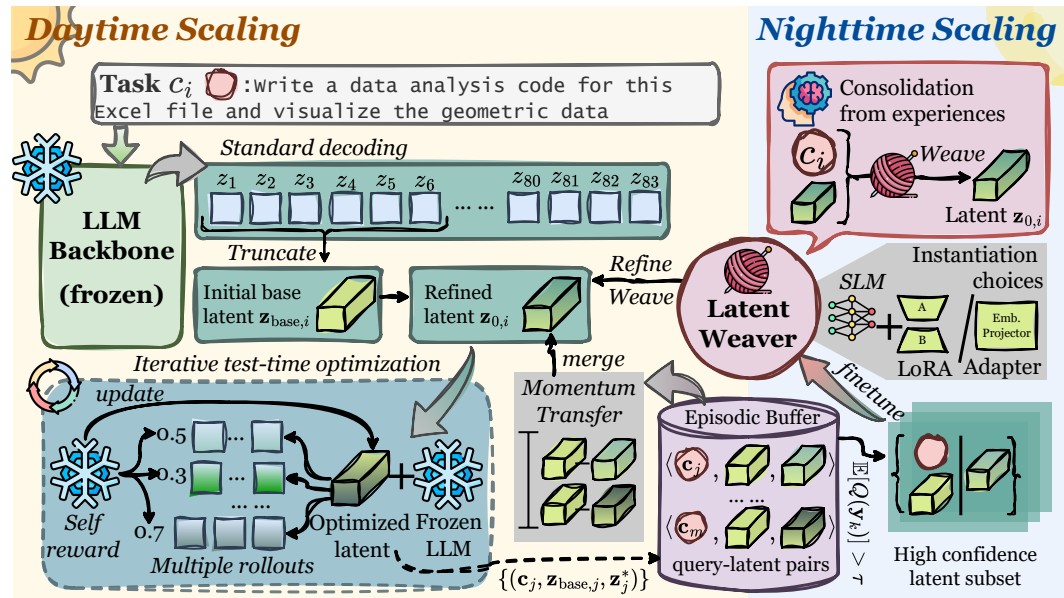

Figure 2: The overview of our proposed `LatentEvolve`.

prompts, $\mathbf{z}_{\text{task}}$, applied across all instances (Xiao et al., 2023; Choi et al., 2023), or performs *query-specific* optimization to find a bespoke latent path $\mathbf{z}_i$ for each individual query $\mathbf{c}_i$ (Li et al., 2025a; Peng et al., 2024; Xu et al., 2025c; Sun et al., 2025). Whatever the granularity is, most of these practices are self-contained, *i.e.*, do not rapidly learn or evolve from one instance to the next, thereby incapable of on-the-fly adaptation based on cumulative experience.

## 4 METHODOLOGY

`LatentEvolve` unfolds as a dual-phase evolving process that enables LLMs to adapt and self-improve at test time. First, we introduce *daytime test-time scaling* (▷ Section 4.1), which performs fast, instance-specific adaptation guided by weighted momentum transfer. Then, *nighttime deliberative consolidation* (▷ Section 4.2) integrates these episodic traces into a compact parametric prior through the latent weaver. Finally, *dual-phase evolving scaling* (▷ Section 4.3) ties the two phases into a recurrent cycle, ensuring continual interleaved evolution in latent space.

### 4.1 DAYTIME TEST-TIME OPTIMIZATION

The *Daytime Scaling* is designed for fast, on-the-fly adaptation, mirroring the brain's ability to rapidly recall specific past experiences to navigate a present challenge. This process unfolds in three key stages for each incoming query: retrieving relevant memories, constructing an informed initial latent sequence, and refining it through self-guided optimization.

**Associative Retrieval.** Inspired by the function of episodic memory in cognitive science, `LatentEvolve` maintains an **episodic buffer**, $\mathcal{M}$, which serves as a dynamic archive of specific, high-quality test-time scaling experiences. Each entry is a triplet $(\mathbf{e}_{\mathbf{c}_j}, \mathbf{z}_{\text{base},j}, \mathbf{z}_j^*)$, storing a previous query's context embedding $\mathbf{e}_{\mathbf{c}_j}$, its initial latent sequence $\mathbf{z}_{\text{base},j}$, and its refined latent sequence $\mathbf{z}_j^*$.

Upon receiving a new query, which we define as the input prompt $\mathbf{c}_i$, we first compute its semantic embedding $\mathbf{e}_{\mathbf{c}_i}$ using the frozen LLM's final hidden state. We then perform a similarity search to retrieve a neighborhood of the top-$k$ most relevant experiences from the buffer:

$$\mathcal{N}_k(\mathbf{c}_i) = \text{Top-k}_j\{(\mathbf{e}_{\mathbf{c}_j}, \mathbf{z}_{\text{base},j}, \mathbf{z}_j^*) \in \mathcal{M}\}, \quad \text{based on similarity } S(\mathbf{e}_{\mathbf{c}_i}, \mathbf{e}_{\mathbf{c}_j}), \quad (3)$$

where $S(\cdot, \cdot)$ is instantiated via cosine similarity. This allows the upcoming test-time optimization to benefit from a small, highly relevant subset of its past experiences.

**Informed Latent Initialization.** A well-informed starting point can substantially improve both the efficiency and quality of reasoning. For each query $\mathbf{c}_i$, we first derive a base initialization $\mathbf{z}_{\text{base},i}$

via an initial Chain-of-Thought (CoT) decoding, taking the prefix of the resulting latent sequence:

$$\mathbf{z}_{\text{base},i} = H_{\boldsymbol{\theta}}(\mathbf{c}_i)_{1:L'} \tag{4}$$

where $H_{\boldsymbol{\theta}}(\mathbf{c}_i)$ denotes the full latent sequence produced by $\pi_{\boldsymbol{\theta}}$ under greedy decoding, and the subscript $1 : L'$ selects the first $L'$ latent vectors. This base state serves as a preliminary reasoning trajectory, which can be further refined using the retrieved neighborhood $\mathcal{N}_k(\mathbf{c}_i)$ to form a superior initialization $\mathbf{z}_{0,i}$. A naive approach might be to simply average the retrieved final latent sequences $\mathbf{z}_j^*$, but this can be misleading as different queries may yield conflicting patterns. Instead, we follow a more intuitive principle: it is not the final latent states that matter most, but the journey from the initial $\mathbf{z}_{\text{base},j}$ to the refined $\mathbf{z}_j^*$. We capture this journey as the optimization "momentum", $\Delta \mathbf{z}_j = \mathbf{z}_j^* - \mathbf{z}_{\text{base},j}$, and introduce **weighted momentum transfer**: By aggregating these momenta weightedly, we guide $\mathbf{z}_{\text{base},i}$ toward regions of the latent space that have been fruitful in the past:

$$\mathbf{z}_{0,i} = \mathbf{z}_{\text{base},i} + \sum_{j \in \mathcal{N}_k(\mathbf{c}_i)} \alpha_j \Delta \mathbf{z}_j, \quad \text{where} \quad \alpha_j \propto \exp(S(\mathbf{e}_{\mathbf{c}_i}, \mathbf{e}_{\mathbf{c}_j})). \tag{5}$$

In this way, the initialization is gently steered not only toward promising regions but also along trajectories that have historically led to better outputs, allowing reasoning to begin with a well-informed and contextually grounded foundation.

**Self-Supervised Refinement and Archiving.** Although the informed initial state $\mathbf{z}_{0,i}$ offers a promising foundation, it is not tailored to the specific context $\mathbf{c}_i$ and thus requires refinement to enhance reasoning performance. We adopt a self-rewarding strategy, a paradigm broadly validated in prior work (Li et al., 2025a; Yuan et al., 2025; Zuo et al., 2025). Concretely, the LLM $\pi_{\theta}$ serves as its own evaluator by assigning a quality score $Q(\mathbf{y}_k)$ to the output $\mathbf{y}_k$ generated under the guidance of $\mathbf{z}_{0,i}$ (see the detailed implementation of $Q(\cdot)$ in Appendix B.1). The latent sequence is then iteratively refined through gradient ascent with respect to this self-supervised signal. The gradient of $J(\mathbf{z}_k)$ is estimated via policy gradient (Williams, 1992) as:

$$\nabla_{\mathbf{z}_k} J(\mathbf{z}_k) = \nabla_{\mathbf{z}_k} \mathbb{E}_{\mathbf{y} \sim p(\mathbf{y}|\mathbf{c}_i, \mathbf{z}_k; \boldsymbol{\theta})}[Q(\mathbf{y})] \approx \frac{1}{M} \sum_{m=1}^{M} Q(\mathbf{y}^{(m)}) \nabla_{\mathbf{z}_k} \log p(\mathbf{y}^{(m)} \mid \mathbf{c}_i, \mathbf{z}_k; \boldsymbol{\theta}), \tag{6}$$

where $\{\mathbf{y}^{(m)}\}_{m=1}^{M}$ are samples drawn from $p(\cdot \mid \mathbf{c}_i, \mathbf{z}_k; \boldsymbol{\theta})$ by $M$ times. Accordingly, the latent state is iteratively updated as $\mathbf{z}_{k+1} \leftarrow \mathbf{z}_k + \eta \nabla_{\mathbf{z}_k} J(\mathbf{z}_k)$, where $\eta$ is the learning rate. The refinement terminates either after $K$ iterations or once $\mathbb{E}[Q(\mathbf{y}_k)]$ has failed to increase for three successive rounds, yielding the final latent state $\mathbf{z}_i^*$, under whose guidance $\pi_{\theta}$ produces the ultimate output $\mathbf{y}$. The triplet $(\mathbf{e}_{\mathbf{c}_i}, \mathbf{z}_{\text{base},i}, \mathbf{z}_i^*)$ is archived into $\mathcal{M}$ whenever $\mathbb{E}[Q(\mathbf{y}_k)]$ exceeds a predefined threshold $\tau$ (see detailed process in Appendix B.2). Thus, the preservation of high-confidence experiences deepens the repository from which `LatentEvolve` continually distills its evolving knowledge.

## 4.2 Nighttime Deliberative Consolidation

While the *daytime scaling* excels at rapid, instance-level adaptation, its knowledge remains fragmented within the buffer. To achieve generalizable improvement, these scattered experiences must be integrated into a coherent procedural skill, which is also the purpose of the *nighttime scaling*, analogous to the neocortex's role in consolidating memories into abstract knowledge during sleep.

**Latent Weaver.** To perform this consolidation, we introduce the *latent weaver* $\mathbf{W}_{\psi}$, aimed at distilling the collective wisdom from the episodic buffer. Technically, $\mathbf{W}_{\psi}$ is trained to predict the refined latent sequence $\mathbf{z}_j^*$ conditioned on the context embedding $\mathbf{e}_{\mathbf{c}_j}$ and the base state $\mathbf{z}_{\text{base},j}$, thereby enabling rapid and precise test-time scaling. We instantiate $\mathbf{W}_{\psi}$ via a smaller LLM $\psi$.

**Consolidation through Experience Replay.** Periodically, after the episodic buffer $\mathcal{M}$ has accumulated a sufficient number of high-confidence experiences, the nighttime consolidation is triggered. The experience triplets $\{(\mathbf{e}_{\mathbf{c}_j}, \mathbf{z}_{\text{base},j}, \mathbf{z}_j^*)\}$ from $\mathcal{M}$ are leveraged to update the parameters $\psi$ of the latent weaver. The training objective is to minimize the reconstruction error between the weaver's prediction and the archived optimal latent sequence:

$$\mathcal{L}(\boldsymbol{\psi}) = \mathbb{E}_{(\mathbf{e}_{\mathbf{c}_j}, \mathbf{z}_{\text{base},j}, \mathbf{z}_j^*) \sim \mathcal{M}} \left[ \left\| \mathbf{W}_{\psi}(\mathbf{e}_{\mathbf{c}_j}, \mathbf{z}_{\text{base},j}) - \mathbf{z}_j^* \right\|_2^2 \right], \tag{7}$$

which effectively *weaves* the sparse, episodic optimization experiences into the continuous parametric space of the model. Through such nighttime scaling, $\mathbf{W}_{\psi}$ is imbued with procedural intuition and capable of generating superior initial reasoning paths for subsequent LLM reasoning. In the next section, we illustrate the overall picture of the dual-phase evolving process.

### 4.3 DUAL-PHASE EVOLVING SCALING

In this section, we formally describe the dual-phase evolution process of `LatentEvolve`. The daytime and nighttime mechanisms, though effective in isolation, realize their full potential when embedded in a recurring cycle that mirrors the brain's complementary learning systems: the hippocampus for rapid encoding of episodic traces and the neocortex for gradual schema formation. For each incoming query $c_i$, the latent weaver $\mathbf{W}_{\psi}$ first transforms the base latent state (except in the initial round, when $\mathbf{W}_{\psi}$ remains untrained) to yield a refined $\mathbf{z}'_{\text{base},i}$. This is followed by daytime scaling, which, via momentum transfer, produces $\mathbf{z}_{0,i}$. Iterative refinement then generates the final latent sequence $\mathbf{z}_i^*$ through self-guided optimization, ensuring that each query benefits not only from episodic recall but also from the procedural insights accumulated during prior nighttime consolidations:

$$\mathbf{z}'_{\text{base},i} = \mathbf{W}_{\psi}(\mathbf{e}_{c_i}, \mathbf{z}_{\text{base},i}), \quad \mathbf{z}_i^* = \Phi_{\text{day}}(\mathbf{c}_i, \mathbf{z}'_{\text{base},i}, \mathcal{M}; \boldsymbol{\theta}), \tag{8}$$

where $\Phi_{\text{day}}$ denotes daytime optimization of a given query $\mathbf{c}_i$ under the assistance of $\mathcal{M}$ and $\pi_\theta$, as described in Equation (6). Over time, the episodic buffer $\mathcal{M}$ accumulates triplets of adaptations $\{(\mathbf{e}_{c_j}, \mathbf{z}_{\text{base},j}, \mathbf{z}_j^*)\}$. At periodic intervals (specifically, we set $T = 200$ test-time instances as one cycle), nighttime scaling is invoked to consolidate accumulated experiences by updating $\mathbf{W}_{\psi}$:

$$\mathbf{W}_{\psi} \leftarrow \Phi_{\text{night}}(\mathcal{M}, \mathbf{W}_{\psi}), \tag{9}$$

where $\Phi_{\text{night}}$ denotes experience replay and parametric distillation, as described in Equation (7). The overall evolution is thus expressed as the alternating transformation

$$(\mathcal{M}, \mathbf{W}_{\psi}) \xrightarrow{\Phi_{\text{day}}} \mathcal{M}' \xrightarrow{\Phi_{\text{night}}} (\mathcal{M}', \mathbf{W}'_{\psi}), \tag{10}$$

which continually refreshes the episodic buffer while also imbuing the weaver with generalized procedural knowledge. In other words, as the essential knowledge has already been integrated into the weaver, after each nighttime consolidation, the episodic buffer is cleared to prevent unbounded growth of the memory space. This perpetual cycle of experience and consolidation allows `LatentEvolve`'s reasoning capabilities to self-evolve on the fly, entirely in an **unsupervised manner without reliance on any external labels**.

## 5 EXPERIMENTS

### 5.1 EXPERIMENT SETTING

**Backbones.** To evaluate the generalizability of `LatentEvolve`, we experiment with LLMs from different families and of varying sizes, including `Llama-3.2-3b` (Grattafiori et al., 2024), `Qwen2.5-7b-instruct` (Qwen et al., 2025), `Qwen3-4b-instruct-2507`, `Qwen3-8b` (Yang et al., 2025), and `Gemma-3-12b-it` (Team et al., 2025).

**Benchmarks.** We conduct a comprehensive evaluation of `LatentEvolve` across eight benchmarks from four task domains: ■ *general QA*, MMLU (Hendrycks et al., 2021a); ■ *mathematical reasoning*, including GSM8K (Cobbe et al., 2021), MATH-500 (Hendrycks et al., 2021b), and AIME 2024/2025 (Li et al., 2024); ■ *scientific reasoning*, SciBench (Wang et al., 2024b) and GPQA-Diamond (Rein et al., 2023); ■ *medical reasoning*, JAMA Clinical Challenge (Chen et al., 2025a). Detailed dataset statistics are listed in Appendix B.4.

**Evaluation Setup.** We apply `LatentEvolve` independently to each benchmark's test set, except for AIME24/25 where the test size is limited, on which we evaluate after applying `LatentEvolve` on MATH-500. We set the maximum generation length to 2048 tokens. The small LLM used for latent weaver $\mathbf{W}_{\psi}$ is consistently set as `Qwen-2.5-1.5b`. The dimension $L'$ in Equation (4) is set as 15, the threshold $\tau$ equals 0.5, and the dual-evolution period $T$ is 200. The learning rate $\eta$ is 0.3, the number of iterations $K = 10$, and the sampling times $M$ in Equation (6) is 8. For performance evaluation, we employ *Pass@1* accuracy under a sampling temperature of 0 across all benchmarks.

**Baselines.** We compare against several well-established baselines:
- **Prompting (training-free)**: vanilla model and CoT (Wei et al., 2023);
- **Reinforcement Learning**: (1) self-rewarding methods, including Self-Rewarding (Yuan et al., 2025) and Genius (Xu et al., 2025a), and (2) verifiable reward methods, including GRPO (DeepSeek-AI et al., 2025), Reinforce (Williams, 1992), and Reinforce++ (Hu et al., 2025). The latter three baselines are trained independently on the training split of each dataset and evaluated on the corresponding test split. Owing to the limited size of AIME24/25, models trained on MATH are directly evaluated on these benchmarks. Results on SciBench are omitted for these baselines due to the absence of a dedicated training set.

Table 1: **Performance Comparison** across two LLM backbones (Qwen2.5-7b and Llama3.2-3b), against thirteen baselines and on eight benchmarks. The best and second best results are highlighted and underlined, respectively.

| Method | General QA | Mathematical Reasoning | | | | Sci. Reasoning | | Med. Reasoning |
|---|---|---|---|---|---|---|---|---|
| | MMLU | GSM8K | MATH-500 | AIME24 | AIME25 | SciBench | GPQA | JAMA Clinical |
| *Prompting (training-free)* | | | | | | | | |
| Vanilla Model | 55.30 | 87.72 | 55.80 | 0.00 | 0.00 | 11.27 | 27.78 | 47.72 |
| CoT | 69.10 | 87.04 | 68.80 | 6.67 | 3.33 | 11.99 | 30.81 | 50.96 |
| *Reinforcement Learning* | | | | | | | | |
| Self-Rewarding | 63.10 | 88.30 | 59.62 | 0.00 | 0.00 | 9.36 | 23.65 | 47.07 |
| Genius | 58.30 | 87.93 | 49.57 | 0.00 | 0.00 | 13.60 | 29.31 | 41.78 |
| GRPO | 68.90 | 92.30 | 75.85 | 6.67 | 3.33 | - | 33.60 | 51.62 |
| Reinforce | 63.77 | 92.30 | 76.80 | 6.67 | 6.67 | - | 34.34 | 49.16 |
| Reinforce++ | 65.90 | 92.60 | 75.02 | 13.33 | 6.67 | - | 34.34 | 50.40 |
| *Latent Reasoning* | | | | | | | | |
| Coprocessor | 68.10 | 83.60 | 53.73 | 6.67 | 6.67 | - | 31.88 | 43.70 |
| SoftCoT | 62.30 | 80.13 | 65.80 | 3.33 | 0.00 | - | 28.28 | 49.70 |
| *Test-time Scaling* | | | | | | | | |
| Self-Consistency | 69.80 | 88.62 | 69.40 | 6.67 | 6.67 | 12.13 | 32.32 | 51.62 |
| Self-Refine | 61.40 | 86.33 | 59.32 | 3.33 | 0.00 | 9.36 | 22.65 | 45.64 |
| LatentSeek | 68.50 | 91.58 | 66.20 | 10.00 | 3.33 | 14.45 | 31.31 | 50.40 |
| TTRL | 70.90 | 92.80 | 77.39 | 23.33 | 13.33 | 13.92 | 33.60 | 49.16 |
| LatentEvolve | 72.30 | 92.98 | 77.60 | 23.33 | 10.00 | 19.79 | 34.85 | 52.94 |
| *Prompting (training-free)* | | | | | | | | |
| Vanilla Model | 60.60 | 71.65 | 41.60 | 0.00 | 0.00 | 6.79 | 26.77 | 45.14 |
| CoT | 57.60 | 64.90 | 48.60 | 0.00 | 0.00 | 7.95 | 26.77 | 45.60 |
| *Reinforcement Learning* | | | | | | | | |
| Self-Rewarding | 57.30 | 69.22 | 39.20 | 0.00 | 0.00 | 3.19 | 23.90 | 40.16 |
| Genius | 58.20 | 73.61 | 38.15 | 0.00 | 0.00 | 6.79 | 21.80 | 45.60 |
| GRPO | 62.70 | 75.30 | 50.20 | 3.33 | 0.00 | - | 28.18 | 46.26 |
| Reinforce | 60.60 | 75.02 | 49.60 | 3.33 | 0.00 | - | 24.50 | 45.60 |
| Reinforce++ | 62.70 | 73.61 | 50.20 | 3.33 | 3.33 | - | 26.26 | 44.80 |
| *Latent Reasoning* | | | | | | | | |
| Coprocessor | 61.50 | 70.08 | 44.90 | 0.00 | 0.00 | - | 21.80 | 42.28 |
| SoftCoT | 58.90 | 73.61 | 46.40 | 0.00 | 0.00 | - | 25.25 | 43.35 |
| *Test-time Scaling* | | | | | | | | |
| Self-Consistency | 59.10 | 66.33 | 49.20 | 0.00 | 0.00 | 8.67 | 27.27 | 45.60 |
| Self-Refine | 58.90 | 68.90 | 44.10 | 0.00 | 0.00 | 4.28 | 20.10 | 42.28 |
| LatentSeek | 49.30 | 55.95 | 38.60 | 0.00 | 0.00 | 5.20 | 26.26 | 32.36 |
| TTRL | 62.10 | 75.02 | 51.00 | 3.33 | 6.67 | 8.07 | 28.18 | 44.80 |
| LatentEvolve | 64.30 | 75.51 | 51.20 | 6.67 | 3.33 | 9.39 | 29.29 | 48.44 |

(Left row labels: Qwen2.5-7b for the first block, Llama3.2-3b for the second block.)

- **Latent Reasoning**, including Coprocessor (Liu et al., 2024) and SoftCoT (Xu et al., 2025c).
- **Test-time Scaling** methods, including Self-Consistency (Wang et al., 2023b), Self-refine (Madaan et al., 2023), LatentSeek (Li et al., 2025a), and TTRL (Xiang et al., 2025).

## 5.2 MAIN RESULTS

**Obs. ❶: LatentEvolve performs well across most task domains.** As shown in Table 1, most baselines fail to deliver consistent gains across all benchmark types. LatentSeek and TTRL excel in mathematical reasoning yet fall short in other domains: for instance, LatentSeek with Llama3.2-3b underperforms the vanilla model on MMLU ($-11.3\%$) and experiences a performance drop on SciBench ($-1.59\%$), while TTRL with Qwen2.5-7b yields limited benefit on JAMA Clinical ($+1.44\%$). In contrast, LatentEvolve not only matches or surpasses TTRL in the general QA domain (e.g., on Qwen2.5-7b, MMLU $+6.4\%$) but also achieves superior results in other domains, such as a $+8.52\%$ improvement on SciBench+Qwen2.5-7b.

**Obs. ❷: LatentEvolve generalizes well across LLM backbones.** In contrast to many baselines whose gains across different LLMs are highly inconsistent (*e.g.*, Coprocessor on GPQA yields $+4.1\%$ with Qwen2.5-7b but $-4.97\%$ with Llama3.2-3b), LatentEvolve consistently delivers positive improvements across models of varying scales, as clearly illustrated in Table 2. Notably, its benefits naturally *scale* with model size: on MATH-500, for example, the improvement rises from $9.6\%$ with Llma3.2-3b to $20.8\%$ with Gemma-3-12b.

Table 2: **Performance Comparison** of the vanilla model versus that enhanced with `LatentEvolve` across five LLM backbones. The $\Delta$ row indicates the absolute improvement.

| LLM Backbone | Method | MMLU | GSM8K | MATH-500 | SciBench | GPQA | JAMA Clinical | AIME24 | AIME25 |
|---|---|---|---|---|---|---|---|---|---|
| `Llama3.2-3b` | Vanilla | 60.60 | 71.65 | 41.60 | 6.79 | 26.77 | 45.14 | 0.00 | 0.00 |
| | +LatentEvolve | 64.30 | 75.51 | 51.20 | 9.39 | 29.29 | 48.44 | 6.67 | 3.33 |
| | $\Delta$ | +3.70 | +3.86 | +9.60 | +2.60 | +2.52 | +3.30 | +6.67 | +3.33 |
| `Qwen2.5-7b` | Vanilla | 55.30 | 87.72 | 55.80 | 11.27 | 27.78 | 47.72 | 0.00 | 0.00 |
| | +LatentEvolve | 72.30 | 92.98 | 77.60 | 19.79 | 34.85 | 52.94 | 23.33 | 10.00 |
| | $\Delta$ | +17.00 | +5.26 | +21.80 | +8.52 | +7.07 | +5.22 | +23.33 | +10.00 |
| `Qwen3-4b` | Vanilla | 71.90 | 89.23 | 61.40 | 12.28 | 34.85 | 51.49 | 10.00 | 3.33 |
| | +LatentEvolve | 73.30 | 92.42 | 78.60 | 31.93 | 38.89 | 53.67 | 23.33 | 16.67 |
| | $\Delta$ | +1.40 | +3.19 | +17.20 | +19.65 | +4.04 | +2.18 | +13.33 | +13.34 |
| `Qwen3-8b` | Vanilla | 72.70 | 87.94 | 55.20 | 6.36 | 28.82 | 53.08 | 3.33 | 3.33 |
| | +LatentEvolve | 78.80 | 90.45 | 73.80 | 10.83 | 32.82 | 54.60 | 26.67 | 23.33 |
| | $\Delta$ | +6.10 | +2.51 | +18.60 | +4.47 | +4.00 | +1.52 | +23.33 | +20.00 |
| `Gemma-3-12b` | Vanilla | 65.80 | 89.23 | 57.40 | 10.84 | 33.33 | 49.50 | 0.00 | 10.00 |
| | +LatentEvolve | 73.90 | 91.89 | 78.20 | 18.93 | 41.92 | 55.06 | 10.00 | 13.33 |
| | $\Delta$ | +8.10 | +2.66 | +20.80 | +8.09 | +8.59 | +5.56 | +10.00 | +3.33 |

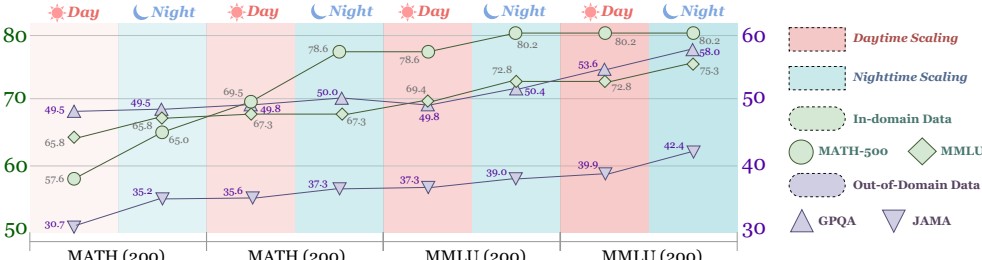

Figure 3: The evolving dynamics of `LatentEvolve` when applied on `Gemma-3-12b` across two in-domain and out-of-domain datasets.

## 5.3 GENERALIZATION AND CONTINUAL LEARNING STUDY

This section investigates the continual learning and generalization capacity of `LatentEvolve`. Figure 3 illustrates performance trajectories when `LatentEvolve`, instantiated with `Gemma-3-12b`, sequentially processes test data from MATH and MMLU. In-domain evaluation is conducted on MATH and MMLU, while out-of-domain evaluation is performed on GPQA and JAMA. Red zones denote evaluations after a daytime scaling step with updated episodic buffer $\mathcal{M}'$, whereas blue zones correspond to evaluations after a nighttime scaling step yielding updated latent weaver $\mathbf{W}'_{\psi}$.

**Obs. ❸: `LatentEvolve` generalizes across domains.** As shown in Figure 3, after two rounds of MATH data, performance improves substantially in-domain ($57.6\% \rightarrow 78.6\%$) while also transferring gains to distinct domains (JAMA $+6.6\%$, MMLU $+1.5\%$). Notably, nighttime scaling proves more conducive to such cross-domain generalization: the first nighttime scaling on MATH increases JAMA by $+4.5\%$, compared to only $+0.4\%$ from daytime scaling. This highlights that nighttime scaling, akin to cortical consolidation in the human brain, integrates experiences into more transferable knowledge, whereas daytime scaling yields more superficial improvements. Moreover, `LatentEvolve` demonstrates strong continual learning ability: after two rounds of MMLU data, `Gemma-3-12b` shows not only no degradation but a slight improvement on MATH ($78.6\% \rightarrow 80.2\%$), highlighting the robustness of `LatentEvolve` in continual adaptation.

## 5.4 FRAMEWORK ANALYASIS

**Sensitivity Analysis.** We conduct a sensitivity analysis of **the parameter** $L'$, which determines the dimensionality of each initialized latent representation $\mathbf{z}_{\text{base},i}$. As shown in Figure 4 (*Left*), both MATH and SciBench exhibit similar patterns: as $L'$ increases from 10 to 50, performance first improves and then declines, with the best results attained at $L' = 30$ (MATH 80.3%, SciBench 33.4%). A plausible explanation is that too few dimensions cannot adequately encode historical optimization experience, while excessively many dimensions introduce additional parameters that

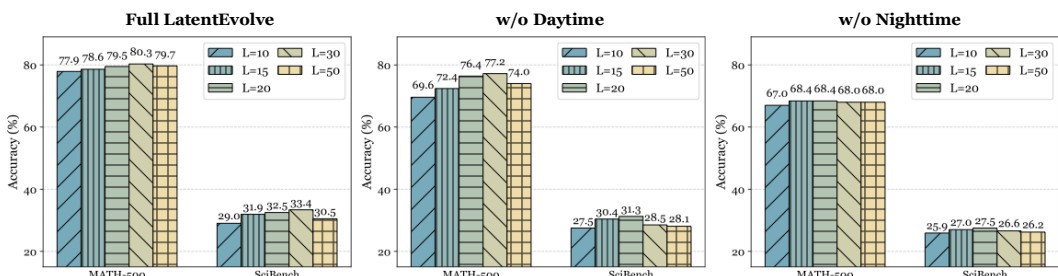

Figure 4: The ablation study and parameter sensitivity analysis of `LatentEvolve`.

may hinder the self-supervised refinement process during daytime scaling (Equation (6)). Analysis with more parameters (*e.g.*, **the evolution period** $T$) is provided in Appendix B.6.

**Ablation Study.** We investigate two variants of `LatentEvolve`: *w/o Daytime* and *w/o Nighttime*. The former removes Equation (5), meaning that each latent optimization proceeds without retrieving prior experience, and the latter omits the update of the latent weaver as described in Equation (7). As shown in Figure 4, removing either daytime or nighttime scaling leads to a substantial performance drop compared with the full `LatentEvolve`, with the effect of *w/o Nighttime* being more pronounced. At $L' = 30$, *w/o Daytime* reduces performance on SciBench from 33.4% to 28.5% ($-4.9\%$), while *w/o Nighttime* results in a larger decline of 6.8%. This observation aligns with the findings in Section 5.3. Overall, both components of `LatentEvolve` are indispensable, mutually reinforcing, and jointly critical to its effectiveness.

Table 3: Case Studies of `LatentEvolve`. The questions (from SciBench and GPQA), the answers, and the `LatentEvolve` generations are listed. The peculiar reasoning tokens are highlighted in blue.

---

**QUESTION:** Express the van der Waals parameters b=0.0226 dm$^3$ mol$^{-1}$ in SI base units.
**ANSWER:** 0.0000226
**LatentEvlove:** To convert parameter b=0.0226 dm$^3$ mol$^{-1}$ in SI base units. Start unit-conv dm3-to-m3 step one. Identify 1 dm equal 0.1 m. So, 1 dm$^3$ is 0.001 m$^3$. Then apply factor-multiplied 0.0226. [...] The final value in SI base units is 2.26 * 10$^{-5}$ m$^3$ mol$^{-1}$.

---

**QUESTION:** Which of the following physical theories never requires regularization at high energies? A. Quantum Chromodynamics B. Superstring Theory C. Quantum Electrodynamics D. Classical Electrodynamics
**ANSWER:** B
**LatentEvlove:** Which physical theories never never required regularization at high energies. Need-no-regularization-scan start. QCD, QED, needs. Classical Electrodynamics, not quantum, so no need. Superstring Theory, this theory no require regularization itself. [...] Based on the above analysis, the theory that never requires regularization at high energies is Superstring Theory. The correct answer is B.

---

**Case Study.** To gain insight into `LatentEvolve`'s latent optimization, we qualitatively analyzed its outputs following (Li et al., 2025a). As shown in Table 3, we observed distinctive reasoning tokens and patterns, including fragmented internal commands (Need-no-regularization-scan start), lexical repetition (never never), and unconventional grammar (no require regularization itself). Despite these peculiar tokens, `LatentEvolve` consistently produces correct answers, suggesting it steers the LLM along more machine-native, efficient reasoning trajectories within latent space. Table 5 further shows that, relative to vanilla CoT, `LatentEvolve` concludes the reasoning with fewer decoding tokens.

## 6 CONCLUSION

In this work, we proposed `LatentEvolve`, a self-evolving latent test-time scaling framework inspired by complementary learning systems. By alternating *daytime scaling* for fast episodic adaptation with *nighttime scaling* for slow procedural consolidation, our approach enables LLMs to accumulate and refine experiential knowledge during inference without external supervision. Experiments across eight benchmarks and five model backbones show that `LatentEvolve` surpasses state-of-the-art TTS methods (*e.g.*, TTRL, LatentSeek), transfers effectively across tasks, and exhibits steady continual learning ability. Broadly, our work points toward a new path where LLMs not only *scale at test time*, but also *evolve through it*, bringing them closer to the adaptive and accumulative intelligence seen in human cognition.

## ETHICS STATEMENT

This work introduces a self-evolving latent test-time scaling framework designed to enhance LLM' adaptability and accumulation of knowledge during inference. Our research is conducted entirely within the scope of standard academic benchmarks, including general reasoning, mathematical problem-solving, and scientific question answering, without deploying models in real-world interactive or decision-making scenarios. As such, the methods and experiments presented here do not raise direct ethical concerns.

## REPRODUCIBILITY STATEMENT

To facilitate the reproducibility of this work, we provide an anonymous link to our source codes in the abstract, detail the parameter settings in Section 5.1, and include the evaluation prompt in Appendices B.1 and B.3.

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

## A USE OF LARGE LANGUAGE MODELS

In preparing this work, we made limited use of large language models (LLMs) as auxiliary tools. Specifically, LLMs were employed to polish the language of early drafts, to assist with literature exploration, and to support information retrieval.

## B METHODOLOGY DETAILS

### B.1 SELF-SUPERVISED LATENT REFINEMENT

The self-rewarding function $Q(\mathbf{y})$ in Equation (6) is formally defined as a weighted aggregation of numerical scores assigned by the LLM to distinct evaluation criteria. Following standard practices from (Lifshitz et al., 2025; Li et al., 2025a), for each candidate output $\mathbf{y}$, the LLM produces normalized scores $s_{\text{ans}}(\mathbf{y}), s_{\text{comp}}(\mathbf{y}), s_{\text{calc}}(\mathbf{y}), s_{\text{form}}(\mathbf{y}), s_{\text{clar}}(\mathbf{y}) \in [0, 1]$, corresponding respectively to (i) correctness of the final answer, (ii) accuracy of problem comprehension, (iii) validity of numerical calculations, (iv) conformity of the answer format to task requirements, and (v) provision of a clear and explicit answer. The overall reward is then computed as

$$Q(\mathbf{y}) = \frac{1}{7}\Big( s_{\text{ans}}(\mathbf{y}) + s_{\text{comp}}(\mathbf{y}) + s_{\text{calc}}(\mathbf{y}) + 2\, s_{\text{form}}(\mathbf{y}) + 2\, s_{\text{clar}}(\mathbf{y}) \Big), \tag{11}$$

where the weighting scheme $1 : 1 : 1 : 2 : 2$ reflects the relative importance of the criteria, placing greater emphasis on answer format fidelity and clarity of presentation.

---

**Prompt for (i) correctness of the final answer**

```
prompt_s_ans = f"""
INSTRUCTIONS:
Your task is to determine the correctness of the final answer within the PROPOSED
    SOLUTION.
Critically verify the final answer against the TASK DESCRIPTION and the reasoning steps
    provided in the PROPOSED SOLUTION.
Do NOT use external knowledge. Focus only on internal consistency and accuracy based on
    the given problem.

Your response must strictly follow the required format:
SCORE: [0.0-1.0]
(0.0 = completely incorrect, 1.0 = perfectly correct)

TASK DESCRIPTION:
[TASK_DESCRIPTION]

PROPOSED SOLUTION:
[PROPOSED_SOLUTION]
"""
```

---

**Prompt for (ii) accuracy of problem comprehension**

```
prompt_s_comp = f"""
INSTRUCTIONS:
Your task is to evaluate the PROPOSED SOLUTION's understanding of the TASK DESCRIPTION.
Identify all explicit and implicit constraints, conditions, and specific requests in
    the TASK DESCRIPTION.
Assess how accurately and comprehensively the PROPOSED SOLUTION addressed these
    elements, demonstrating full comprehension.

Your response must strictly follow the required format:
SCORE: [0.0-1.0]
(0.0 = no comprehension, 1.0 = full and accurate comprehension)

TASK DESCRIPTION:
[TASK_DESCRIPTION]

PROPOSED SOLUTION:
[PROPOSED_SOLUTION]
"""
```



### Prompt for (iii) validity of numerical calculations

```
prompt_s_calc = f"""
INSTRUCTIONS:
Your task is to verify the validity of all numerical calculations and logical steps
    within the PROPOSED SOLUTION.
For each calculation or logical transition, independently recompute or re-evaluate it.
If any numerical or logical discrepancy is found, it indicates an error.

Your response must strictly follow the required format:
SCORE: [0.0-1.0]
(0.0 = many errors, 1.0 = all calculations and logical steps are valid)

TASK DESCRIPTION:
[TASK_DESCRIPTION]

PROPOSED SOLUTION:
[PROPOSED_SOLUTION]
"""
```





### Prompt for (iv) conformity of the answer format to task requirements

```
prompt_s_form = f"""
INSTRUCTIONS:
Your task is to assess if the PROPOSED SOLUTION conforms to the expected output format
    requirements.
Consider if specific units are used, if the answer is structured as implicitly or
    explicitly requested (e.g., numeric only, step-by-step, \\boxed{} formatting), and
     if all parts of the response are appropriately presented.

Your response must strictly follow the required format:
SCORE: [0.0-1.0]
(0.0 = completely incorrect format, 1.0 = perfectly formatted)

TASK DESCRIPTION:
[TASK_DESCRIPTION]

PROPOSED SOLUTION:
[PROPOSED_SOLUTION]
"""
```





### Prompt for (v) provision of a clear and explicit answer

```
prompt_s_clar = f"""
INSTRUCTIONS:
Your task is to evaluate the clarity and explicitness of the PROPOSED SOLUTION.
Assess if the reasoning is easy to follow, unambiguous, and if all necessary steps and
    explanations are provided without missing information.
Consider the overall readability and conciseness.

Your response must strictly follow the required format:
SCORE: [0.0-1.0]
(0.0 = very unclear/implicit, 1.0 = exceptionally clear and explicit)

TASK DESCRIPTION:
[TASK_DESCRIPTION]

PROPOSED SOLUTION:
[PROPOSED_SOLUTION]
"""
```



## B.2 NIGHTTIME CONSOLIDATION

When preparing training data for the latent weaver, not all encountered latent representations within a given cycle are equally valuable or should be retained. Recall that the rationale behind self-supervised refinement is as follows: although the latent state $z_{0,i}$, obtained through weighted momentum transfer, may lie within a promising region of the space, it is not guaranteed to align perfectly with the current context. To address this, we further scale and update it through a self-

rewarding mechanism, ensuring that it becomes optimally adapted to support the query at hand. Therefore, the latent weaver, which provides the initial seeds for reasoning, should itself be trained with relatively high-quality data. Concretely, for a triplet $(\mathbf{e}_{c_i}, \mathbf{z}_{0,i}, \mathbf{z}_i^*)$, we include it into the memory $\mathcal{M}$ if and only if the LLM exhibits sufficient confidence in the associated latent representation. Formally, such confidence is defined as the expected quality score of the final output after iterative refinement. Let $\mathbf{y}_k^{(M)}$ denote the response generated at the last refinement step under rollout $k$ ($k = 1, \ldots, M$), then the confidence measure is given by

$$\mathbb{E}[Q(\mathbf{y}_k^{(M)})] = \frac{1}{M} \sum_{k=1}^{M} Q(\mathbf{y}_k^{(M)}),$$

and the triplet is retained only if

$$\mathbb{E}[Q(\mathbf{y}_k^{(M)})] \geq \tau,$$

where $Q(\mathbf{y}_k^{(M)}) \in [0, 1]$ denotes the numerical score assigned to the generated response according to task-specific evaluation criteria, and $\tau$ is a tunable threshold that governs the admission of latent experiences. We set $\tau = 0.5$ across all experiments.

### B.3 EVALUATION

The evaluation prompts used by `LatentEvolve` for datasets requiring numerical answers (including GSM8K, MATH, AIME 2024/2025, and SciBench) and for multiple-choice datasets (including MMLU, SciBench, JAMA, and GPQA) are summarized in Table 4.

Table 4: Evaluation prompts for `LatentEvolve` and other baselines.

| |
|---|
| **Numerical-answer evaluation prompt:** {Question Description}. Please reason step by step, and enclose your final answer within \\boxed{}. |
| **Multiple-choice evaluation prompt:** {Question Description}. Please select the correct option (A, B, C, or D) to answer the question. Your response should be formatted as follows: The correct answer is {your answer option letter here}. |

### B.4 DATASET DETAILS

This section provides the fine-grained statistics of each dataset:

- **MMLU** (Hendrycks et al., 2021a): following prior practice (Yue et al., 2025), we sample 1000 instances.
- **MATH** (Hendrycks et al., 2021b): we adopt the standard MATH-500 subset.
- **GSM8K** (Cobbe et al., 2021): we opt for the full test set (1319 problems).
- **GPQA** (Rein et al., 2023): we employ the GPQA-Diamond subset containing 198 graduate-level questions of elevated difficulty.
- **SciBench** (Wang et al., 2024b): we include all 692 tasks.
- **JAMA Clinical Challenge** (Chen et al., 2025a): comprising questions derived from demanding clinical cases, we adopt all 1511 test items.
- **AIME 2024 and 2025** (Li et al., 2024): each consists of 30 problems.

### B.5 MORE RESULTS

### B.6 SENSITIVITY ANALYSIS

From Table 6, we observe a clear improvement in performance with iterative refinement under moderate evolution intervals. For example, when $T = 200$, accuracy steadily increases from 68.3% at Iter 1 to 73.6% at Iter 5, indicating that frequent daytime and nighttime interactions allow the latent representations to be progressively aligned with the query context. A similar trend is seen for $T = 300$ and $T = 500$, although the performance gains diminish as the interval grows. In contrast,

Table 5: **Algorithmic Statistics** of `LatentEvolve`. We report the ratio of answer length (when equipped with `LatentEvolve`) to vanilla CoT length, under three settings (full `LatentEvolve`, *w/o Daytime* and *w/o Nighttime*).

| | **Model** | Qwen2.5 7b | Qwen3 8b | Qwen3 4b | Llama3.2 3b | **Avg** |
|---|---|---|---|---|---|---|
| **GSM8K** | `LatentEvolve` | 0.89 | 0.92 | 0.91 | 0.96 | 0.92 |
| | *w/o Daytime* | 0.92 | 0.92 | 0.90 | 0.97 | 0.93 |
| | *w/o Nighttime* | 0.98 | 0.95 | 0.94 | 0.94 | 0.95 |
| **MATH-500** | `LatentEvolve` | 0.91 | 0.93 | 0.95 | 0.97 | 0.94 |
| | *w/o Daytime* | 0.93 | 0.93 | 1.01 | 0.99 | 0.97 |
| | *w/o Nighttime* | 1.01 | 0.97 | 0.97 | 0.98 | 0.98 |
| **SciBench** | `LatentEvolve` | 0.93 | 0.95 | 0.94 | 0.98 | 0.95 |
| | *w/o Daytime* | 0.94 | 0.98 | 0.95 | 1.04 | 0.98 |
| | *w/o Nighttime* | 0.99 | 0.95 | 0.92 | 1.02 | 0.97 |

when $T = 1000$, the evolution becomes excessively coarse-grained, essentially reducing the process to a single daytime and nighttime interaction. This hinders the model's ability to perform gradual refinement, thereby limiting the benefits of the proposed dual-phase evolution. These results suggest that overly sparse consolidation undermines the advantages of iterative scaling, while moderate intervals strike a balance between stability and adaptability.

Table 6: Results across different iterations with different $T$ on the MMLU dataset ($T$ denotes the evolution interval controlling the frequency of daytime and nighttime interactions). Smaller $T$ values allow more iterative refinements within the same training budget, leading to smoother convergence, while larger $T$ values reduce the number of available iterations, coarsening the evolution process.

| | **Iter 1** | **Iter 2** | **Iter 3** | **Iter 4** | **Iter 5** |
|---|---|---|---|---|---|
| **200** | 68.3 | 70.2 | 72.8 | 73.3 | 73.6 |
| **300** | 70.2 | 71.9 | 73.3 | 73.3 | - |
| **500** | 72.1 | 73.9 | - | - | - |
| **1000** | 72.8 | - | - | - | - |

# C ADDITIONAL RESULTS

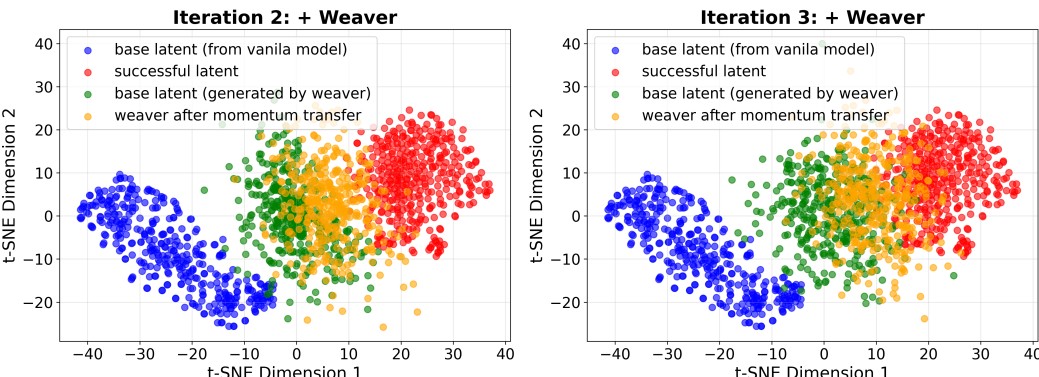

Figure 5: The visualizations of (1) base latent $\mathbf{z}_{base}$, (2) the weaver-refined latent embeddings $\mathbf{z}'_{base}$, (3) the latent embeddings after momentum trasfer $\mathbf{z}_0$, as well as (4) the final latent vectors $\mathbf{z}^*$ (we denote those that lead to the final successful reasoning as *success latent*).

## C.1 VISUALIZATION OF LATENT-REASONING EVOLUTION

Recall that in LatentEvolve, for each task query we first obtain the base latent vector $\mathbf{z}_{\text{base}}$. The nighttime weaver then produces the refined latent $\mathbf{z}'_{\text{base}}$, which is transformed into $\mathbf{z}_0$ after momentum transfer in Equation (5), and after daytime optimization we obtain the final latent $\mathbf{z}^*$, which we refer to as the *success latent* when the resulting trajectory is correct. We apply t-SNE to visualize the distributions of these three latent clusters on SciBench.

As shown in Figure 5, the base latents remain far from the success-latent cluster. Across subsequent iterations, the refined latents produced by the nighttime weaver progressively move toward the region associated with successful reasoning. This indicates that repeated nighttime consolidation enables the weaver to supply latent initializations that lie closer to successful inference regions. The momentum transfer process also steers the latent representations closer to the success region, which validates the effectiveness of momentum-based experience transfer.

## C.2 QUANTITATIVE COMPARISON OF LATENT-CLUSTER DISTANCES

To complement the qualitative visualization above, we compute the Euclidean distances between the centroids of the latent clusters at different iterations. The results are summarized below.

Table 7: Euclidean distances between latent-cluster centroids on `Llama-3.2-3B` + JAMA. Iterations correspond to successive daytime–nighttime cycles.

| Llama-3.2-3B + JAMA | Iter 1 | Iter 2 | Iter 3 |
|---|---|---|---|
| base latent ↔ success latent | 2317.401 | 2466.180 | 2289.177 |
| refined latent ↔ success latent | – | 1094.228 | 834.102 |

Table 8: Euclidean distances between latent-cluster centroids on `Qwen2.5-7B` + MATH.

| Qwen2.5-7B + MATH | Iter 1 | Iter 2 |
|---|---|---|
| base latent ↔ success latent | 1343.296 | 1308.065 |
| refined latent ↔ success latent | – | 955.109 |

Across both settings, the refined latents consistently move closer to the success-latent region as daytime and nighttime phases alternate, while the base latents remain substantially farther away. These quantitative findings support the conclusion that nighttime consolidation provides a significantly improved initialization point for downstream reasoning.

## C.3 EFFICIENCY ANALYSIS

To further clarify the computational cost of **LatentEvolve** during nighttime optimization, we report detailed measurements across representative model–dataset configurations. Table 9 and Table 10 summarize the nighttime GPU hours, daytime wall-clock time, number of nighttime training samples, and the corresponding performance across iterations of latent evolution.

Table 9: Nighttime computational cost and performance across iterations for Llama-3.2-3B on JAMA. Each iteration uses $T = 200$ test samples.

| Metrics | Iter 1 | Iter 2 | Iter 3 |
|---|---|---|---|
| Nighttime GPU hours (h) | 1.23 | 1.11 | 1.33 |
| Daytime wall-clock time (h) | 2.5 | 2.5 | 2.4 |
| Nighttime training samples | 127 | 133 | 139 |
| Performance (Acc.) | 46.8% | 47.2% | 47.9% |

Table 10: Nighttime computational cost and performance across iterations for Qwen2.5-7B on MATH. Each iteration uses $T = 250$ test samples.

| Metrics | Iter 1 | Iter 2 |
|---|---|---|
| Nighttime GPU hours (h) | 1.3h | 1.3h |
| Daytime wall-clock time (h) | 2.07 | 2.12 |
| Nighttime training samples | 188 | 213 |
| Performance (Acc.) | 74.2% | 77.8% |

Overall, the nighttime computational overhead of `LatentEvolve` remains modest across models and domains, while each iteration consistently yields measurable performance gains in a fully unsupervised setting. These results demonstrate that `LatentEvolve` provides an efficient and practical optimization procedure without imposing substantial additional resource burden.

## C.4 RETRIEVAL SIMILARITY METRIC

To assess the influence of the retrieval module on overall system performance, we further evaluate several alternative similarity measures beyond the default cosine similarity. Specifically, we compare cosine similarity, dot product, Pearson correlation, and Euclidean distance within the latent retrieval component. As summarized in Table 11, `LatentEvolve` exhibits limited sensitivity to the choice of similarity function: all variants achieve comparable performance, and cosine similarity remains a strong and robust default across both settings.

Table 11: Ablation on retrieval similarity metrics.

| Metric | MATH + `Qwen2.5-7B` | JAMA + `Llama-3.2-3B` |
|---|---|---|
| Cosine similarity | 77.8 | 48.4 |
| Dot product | 77.4 | 47.5 |
| Pearson correlation | 76.7 | 48.0 |
| Euclidean distance | 75.9 | 47.1 |

## C.5 ANALYSIS OF THRESHOLD $\tau$

To examine the effect of the confidence threshold $\tau$ used for selecting nighttime training samples, we perform a sensitivity study across a range of values. Table 12 reports performance under $\tau \in \{0.3, 0.5, 0.7, 0.9\}$ for three representative model–data pairs. We observe that both overly small and overly large thresholds negatively impact performance: a small $\tau$ admits many low-confidence or noisy samples into nighttime consolidation, while a large $\tau$ prunes too aggressively and thus reduces the amount of useful training signal available to the latent weaver. In practice, moderate values ($\tau = 0.5$ or $0.7$) yield consistently stable behavior, and our default choice of $\tau = 0.5$ provides a robust and broadly effective setting across all evaluated configurations.

Table 12: Sensitivity of LatentEvolve to the confidence threshold $\tau$.

| $\tau$ | 0.3 | 0.5 | 0.7 | 0.9 |
|---|---|---|---|---|
| `Qwen2.5-7B` + MATH | 76.0 | 77.6 | 76.5 | 74.2 |
| `Llama-3.2-3B` + JAMA | 46.2 | 48.4 | 48.8 | 47.1 |
| `Qwen3-8B` + GPQA | 32.6 | 32.8 | 31.6 | 29.2 |

## C.6 AIME AVG@32 RESULTS

Table 13: AIME datasets supplementary results. Note that for GRPO, Reinforce baselines, the models are first trained on the MATH training split, and the resulting checkpoints are then used for inference on AIME datasets.

| Method | Qwen2.5-7B | | Llama3.2-3B | |
|---|---|---|---|---|
| | AIME24 (Avg@32) | AIME25 (Avg@32) | AIME24 (Avg@32) | AIME25 (Avg@32) |
| Vanilla Model | 11.84 | 8.31 | 1.93 | 1.04 |
| CoT | 11.95 | 8.92 | 2.03 | 0.94 |
| Self-Rewarding | 6.25 | 9.89 | 4.79 | 2.97 |
| Genius | 6.82 | 9.11 | 2.92 | 2.50 |
| GRPO | 18.49 | 15.10 | 6.20 | 5.89 |
| Reinforce | 17.70 | 14.27 | 5.57 | 5.52 |
| Reinforce++ | 18.75 | 14.94 | 5.21 | 5.36 |
| Coprocessor | 14.16 | 10.26 | 1.25 | 0.00 |
| SoftCoT | 10.78 | 8.17 | 1.82 | 1.61 |
| Self-Consistency | 14.94 | 10.20 | 3.59 | 3.02 |
| Self-Refine | 13.38 | 8.64 | 2.55 | 2.34 |
| LatentSeek | 14.06 | 15.41 | 6.15 | 4.74 |
| TTRL | 23.33 | 16.71 | 8.65 | 6.46 |
| LatentEvolve | 21.56 | 18.37 | 8.85 | 5.92 |

