# OpenReview forum: "LatentEvolve: Self-Evolving Test-Time Scaling in Latent Space"
_ICLR.cc/2026/Conference — Submitted to ICLR 2026_

### Official Review · Reviewer_HDXy · 2025-10-24

**Soundness:** 3
**Presentation:** 3
**Contribution:** 3
**Rating:** 6
**Confidence:** 3

**Summary:**

The paper proposes LatentEvolve, a latent test-time scaling framework that lets an LLM self-evolve across queries without parameter updates to the backbone. This analogy to the human brain’s dual system is very interesting. The experimental results are excellent across eight benchmarks and five models. LatentEvolve often matches or exceeds state-of-the-art TTS baselines (e.g., TTRL, LatentSeek) and shows cross-domain transfer and continual learning dynamics.

**Strengths:**

1. This analogy to the human brain’s dual system is very interesting.
2. The weighted momentum transfer uses change vectors $\Delta z$ from similar tasks—intuitively leveraging “how we improved” rather than “where we ended”.
3. The experiments are very extensive and comprehensive, and demonstrate consistent gains over strong TTS baselines and transfer to out-of-domain datasets, suggesting a path toward continual, unsupervised evolution at inference.
4. The writing is clear and easy to follow.

**Weaknesses:**

1. This method introduces multiple rollout refinement steps, retrieval over a growing buffer, and periodic weaver training for every instance. The paper should quantify total inference FLOPs/tokens vs. baselines under matched compute budgets, and report throughput/latency impacts.

2. While ablations remove Daytime/Nighttime, there is no scaling analysis of the episodic buffer size, retrieval top-$k$, or archive threshold $\tau$. These directly affect quality and cost.

3. Some AIME scores are based on small sets and can be volatile. I suggest reporting mean ± stdev over ≥3 runs in different sampling seeds.

**Questions:**

No

---

> ### Author Response · Authors · 2025-11-20
> **[Part 1/1] Author Response**
>
> We sincerely thank you for your careful comments and thorough understanding of our paper! Here we give point-by-point responses to your comments and describe the revisions we made to address them.
>
>
> ---
> > **`Weakness 1`**: This method introduces multiple rollout refinement steps, retrieval over a growing buffer, and periodic weaver training for every instance. The paper should quantify total inference FLOPs/tokens vs. baselines under matched compute budgets, and report throughput/latency impacts.
>
>
> We are sincerely grateful to the reviewer for this insightful comment. To clarify the computational implications of LatentEvolve, we provide both per-iteration cost statistics and comparisons against baselines under matched wall-clock budgets.
>
> First, we report the time breakdown for one full daytime–nighttime cycle:
>
> *[Llama-3.2-3B + JAMA. “Iter 1” denotes one full cycle of daytime + nighttime scaling, each iteration using $T=200$ test samples (explicitly stated on Line 284).]*
> | Metrics | Iter 1 | Iter 2 | Iter 3 |
> |---------|--------|--------|--------|
> | Nighttime GPU hours (h) | 1.23h | 1.11h | 1.33h |
> | Daytime wall-clock time | 2.5h | 2.5h | 2.4h |
> | Nighttime training samples | 127 | 133 |  139|
> | Performance (Acc.) | 46.8% | 47.2% | 47.9% |
>
>
> These measurements show that the per-iteration consolidation cost remains moderate and does not grow across cycles.
>
>
> Second, we compare LatentEvolve with the vanilla model and the LatentSeek baseline under comparable computational budgets:
>
>
> *[Comparison conducted on Qwen2.5-7b model on a L40 (48GB) GPU.]*
> |Method|MATH||SciBench||
> |-|-|-|-|-|
> ||Wall-clock time|Perf.|Wall-clock time|Perf.|
> |Vanilla|2.5h|55.80|4.3h|11.27
> |LatentSeek|5.8h|66.20|8.1h|14.45
> |LatentEvolve|6.2h|77.60|9.6h|19.79|
>
> Under nearly matched inference budgets, LatentEvolve consistently achieves substantially stronger accuracy. Relative to LatentSeek, it requires only about 7% additional wall-clock time (6.2h vs. 5.8h) but yields a 17.2% relative improvement on MATH and similarly large gains on SciBench.
>
>
> Overall, we respectfully argue that LatentEvolve introduce modest computational overhead while translating into disproportionately large performance improvements, which test-time scaling methods exactly look for.
>
>
> ---
> > **`Weakness 2`**: While ablations remove Daytime/Nighttime, there is no scaling analysis of the episodic buffer size, retrieval top-k, or archive threshold $\tau$. These directly affect quality and cost.
>
>
> We sincerely appreciate the reviewer’s insightful suggestions, which have helped us strengthen the clarity of our manuscript. We address each component in turn:
> * **Episodic buffer size**
>   As already detailed in Appendix B.6, we conduct a sensitivity study on the episodic buffer size $T$, evaluating $T \in \\{200, 300, 500, 1000\\}$. We observe that $T = 200$ provides a favorable trade-off between update frequency and stability, and we adopt this setting across all datasets without requiring additional tuning.
> * **Retrieval top-$k$**
>   We use a unified setting of $k = 15$ in all main experiments. Following the reviewer’s suggestion, we additionally report a sensitivity analysis below. Retrieving too few latents reduces the effectiveness of momentum transfer during daytime scaling, whereas larger $k$ values yield diminishing returns. Overall, $k = 15$ offers strong and stable performance without the need for frequent adjustment.
>
> *[Experiments conducted with Qwen2.5-7b.]*
> |k|5|10|15|20|30|
> |-|-|-|-|-|-|
> |MATH| 75.6|76.2|77.6|76.8|77.3|
> |SciBench|17.48|19.41|19.76|19.65|20.30|
>
> - **Threshold $\tau$**: As stated on Line 313 of our manuscript, we fix $\tau = 0.5$ across all experiments. To further validate that this choice is generally effective and does not require heavy tuning, we conducted a sensitivity analysis:
>
> | $\tau$ | 0.3 | 0.5 | 0.7 | 0.9 |
> |--------|-----|-----|-----|-----|
> | Qwen2.5-7B + MATH | 76.0 | 77.6 | 76.5 | 74.2 |
> | Llama-3.2-3B + JAMA | 46.2 | 48.4 | 48.8 | 47.1 |
> | Qwen3-8B + GPQA | 32.6 | 32.8 | 31.6 | 29.2 |
>
> We observe that both overly small and overly large values of $\tau$ lead to performance degradation: a small $\tau$ admits many low-confidence or noisy samples into nighttime consolidation, whereas a large $\tau$ filters out too many samples, substantially reducing the effective training signal for the latent weaver. Overall, values in the range $\tau \in \\{0.5, 0.7\\}$ yield stable performance, and our choice of $\tau = 0.5$ serves as a robust, broadly applicable default.
>
> ---
>
> > **`Weakness 3`**: Some AIME scores are based on small sets and can be volatile. I suggest reporting mean ± stdev over ≥3 runs in different sampling seeds.
>
>
> We sincerely appreciate your guidance! In response, we have added the corresponding AIME 2024/2025 Average@32 results in Appendix C.6. The new evaluation confirms that LatentEvolve continues to demonstrate strong performance across benchmarks.

---

> ### Author Response · Authors · 2025-11-25
> **Thank you & Looking forward to further discussion**
>
> Dear Reviewer **HDXy**,
>
> We deeply appreciate your dedication to engaging in author–reviewer discussions. Recognizing the existing discrepancies in the understanding of our manuscript, we have outlined your key concerns and our responses for enhanced communication:
>
> 1. **Regarding computational overhead `Weakness 1`**: We clarified the full daytime–nighttime cost and showed that LatentEvolve introduces only modest additional computation while delivering substantially higher accuracy under matched budgets.
> 2. **Regarding core hyperparameters `Weakness 2`**: We added sensitivity studies for retrieval top-k and the threshold τ and respectfully noted that the analysis for buffer size is already detailed in Appendix B.6.
> 3. **Regarding AIME score stability `Weakness 3`**: We have supplemented the corresponding AIME 2024/2025 Average@32 results in Appendix C.6.
>
> For other issues not mentioned here, please refer to our detailed rebuttal response. We sincerely hope this addresses your concerns! We humbly look forward to further discussion with you.
>
> Warm regards,
> **Authors**

---

> > ### Comment · Reviewer_HDXy · 2025-11-25
> > **Thank you**
> >
> > Thank you for the extensive experimental updates and the clear clarifications provided. They have fully addressed my concerns. I hope to see these valuable additions incorporated into the final camera-ready version.

---

> > > ### Author Response · Authors · 2025-11-26
> > > **Thank you immensely!**
> > >
> > > We would like to sincerely thank you for your inspiring discussion and support of our work. We are more than happy to see that our rebuttal has properly addressed your concerns!

---

### Official Review · Reviewer_8Ufs · 2025-10-26

**Soundness:** 3
**Presentation:** 2
**Contribution:** 2
**Rating:** 4
**Confidence:** 4

**Summary:**

This paper proposes a new framework, called LatentEvolve, for improving test-time scaling (TTS) of large language models (LLMs) by introducing a self-evolving latent-space mechanism. Key contributions include:
Identification of a limitation in existing TTS approaches: most treat each query independently and thus do not accumulate experience across queries to improve future inference.


* Inspiration from the complementary learning systems (CLS) theory (hippocampus + neocortex) to design a dual-phase process:


* Daytime scaling: fast retrieval of prior latent interventions + on-the-fly latent optimization for each incoming query.


* Nighttime scaling: consolidation of past experience into a compact latent “weaver” model that produces better initial latent states for future queries.



* Empirical evaluation across multiple LLM backbones (e.g., Llama-3, Qwen, Gemma) and eight benchmarks spanning general QA, math reasoning, scientific reasoning, medical reasoning. The method shows consistent improvements over strong baselines including LatentSeek, TTRL, etc.

* Evidence of cross-domain generalization (i.e., improvements transfer from math domain to GPQA, JAMA) and continual learning capability (processing new domains sequentially doesn’t degrade performance on previous domains) via the dual-phase mechanism.

**Strengths:**

* The analogy to CLS (fast + slow learning) offers a fresh lens on test-time adaptation / scaling.


* The authors test across several backbones, many dataset types, and compare to a large set of baselines. This breadth increases confidence in generality.


* The idea that the system accumulates experience over time and becomes better aligned with real-world deployment where models repeatedly see new inputs.

**Weaknesses:**

* The daytime latent optimization (via self-rewarding policy gradient) and the nighttime consolidation add overhead. The paper would benefit from more discussion of runtime, cost, scalability (especially for very large models or high query volume).


* It is unclear how sensitive the performance is to the design choices of latent-token length, initialization, buffer size, retrieval similarity metric, and tuning of thresholds (τ), iteration count (K), etc. Some of these hyperparameters may require heavy tuning.


* The episodic buffer grows with experiences; managing retrieval, similarity search, etc might become onerous in production. More details on buffer capacity, retrieval latency, memory demands would strengthen the paper.  Also,  the retrieval process  relies on semantic embedding. If the retrieval process is inaccurate, how does it influence the performance?


* The self-rewarding strategy uses the model itself to score Q(y). There is an inherent risk of circularity or bias: if the model is wrong but confident, it may reinforce the wrong latent sequence. The paper would benefit from ablations or analysis of reward reliability.


* While some cross-domain results are given, one may ask how the system performs when the new queries are far from any prior buffer entries (i.e., the retrieval gives weak plus few relevant experiences). Does the system fallback gracefully or degrade badly?

**Questions:**

* Could you provide detailed statistics on the inference cost (wall-clock time, latency) of daytime scaling + retrieval + latent optimization vs. a vanilla inference pass? In deployment settings, latency is often critical.





* If the system is exposed to a query from a domain completely unseen so far (so retrieval returns low‐similarity neighbours), how does performance degrade? Is there a fallback default initialization?




* Could you share some qualitative examples of latent sequences (zbase → z*), or visualise how the latent space evolves over time (e.g., clustering of queries, drift in latent interventions)? This would help understand what the system is learning.

---

> ### Author Response · Authors · 2025-11-20
> **[Part 1/4] Author Response**
>
> We would like to express our deepest respect for your meticulous review!  In response to your comments, we have carefully prepared a point-by-point reply:
>
> ---
>
> > **`Weakness 1`**: The paper would benefit from more discussion of runtime, cost, scalability (especially for very large models or high query volume).
>
>
> Thank you for this important question! We respectfully provide a detailed breakdown of LatentEvolve's computational cost across different configurations:
>
>
> *[Llama-3.2-3B + JAMA. “Iter 1” denotes one full cycle of daytime + nighttime scaling, each iteration using $T=200$ test samples (explicitly stated on Line 284).]*
> | Metrics | Iter 1 | Iter 2 | Iter 3 |
> |---------|--------|--------|--------|
> | Nighttime GPU hours (h) | 1.23h | 1.11h | 1.33h |
> | Daytime wall-clock time | 2.5h | 2.5h | 2.4h |
> | Nighttime training samples | 127 | 133 |  139|
> | Performance (Acc.) | 46.8% | 47.2% | 47.9% |
>
> *[Qwen2.5-7B + MATH.]*
> | Metrics | Iter 1 | Iter 2 |
> |---------|--------|--------|
> | Nighttime GPU hours (h) | 1.3h | 1.3h |
> | Daytime wall-clock time | 1.8h | 1.8h |
> | Nighttime training samples | 188 | 213 |
> | Performance (Acc.) | 74.2% |77.8%  |
>
> As shown, the nighttime resource requirements for LatentEvolve remain moderate across configurations, while the framework provides substantial performance improvements in a fully unsupervised manner. We sincerely hope this addresses your concerns regarding nighttime computational efficiency.
>
>
>
> ---
>
> > **`Weakness 2`**: It is unclear how sensitive the performance is to the design choices of latent-token length, initialization, buffer size, retrieval similarity metric, and tuning of thresholds (τ), iteration count (K), etc. Some of these hyperparameters may require heavy tuning.
>
>
> We would like to address each of the elements you mentioned and clarify which aspects have already been analyzed in the manuscript, as well as highlight the few that require additional discussion:
>
> - **[Already discussed] Latent token length $L'$**: As detailed in Section 5.4, Figure 4 reports performance across different choices of $L'$. We observe that $L' \in [15, 30]$ consistently yields strong results, indicating that the parameter does not require heavy tuning.
> - **[Already discussed] Initialization**: As specified in Eq. (4), the initialization of base latent vectors is obtained directly by extracting the first $L'$ tokens from standard CoT decoding, providing a simple and well-defined starting point.
> - **[Already discussed] Buffer size**: Appendix B.6 presents an explicit sensitivity study on the episodic buffer size $T$, evaluating $T \in \{200, 300, 500, 1000\}$. We find that $T = 200$ offers a favorable balance and is used across all datasets without requiring extensive tuning.
> - **[Newly added] Retrieval similarity metric**: As stated on Line 212, the retrieval module originally uses cosine similarity, one of the most common choices in representation-based retrieval. Following your valuable suggestion, we additionally conducted an ablation study comparing several alternative metrics. The results are shown below. We observe that LatentEvolve is not highly sensitive to the choice of similarity metric, and cosine similarity remains a strong and robust default.
>
>
> |Metric|MATH + Qwen2.5-7B| Llama-3.2-3B + JAMA|
> |-|-|-|
> |Cosine similairty|77.8|48.4|
> |Dot product|77.4|47.5|
> |Pearson coeficient|76.7|48.0|
> |Euclidean distance|75.9|47.1|
>
> - **[Newly Added] Threshold $\tau$**: As stated on Line 313, we fix $\tau = 0.5$ across all experiments. To confirm that this choice is broadly effective and **does not require extensive tuning**, we performed a sensitivity study as shown below. We can observe that both very small and very large thresholds reduce performance: small values allow low-confidence or noisy samples into nighttime consolidation, whereas large values overly restrict training data for the weaver. Values in the range $\tau \in \{0.5, 0.7\}$ remain stable, and $\tau = 0.5$ serves as a reliable, generally applicable default.
>
>
> | $\tau$ | 0.3 | 0.5 | 0.7 | 0.9 |
> |--------|-----|-----|-----|-----|
> | Qwen2.5-7B + MATH | 76.0 | 77.6 | 76.5 | 74.2 |
> | Llama-3.2-3B + JAMA | 46.2 | 48.4 | 48.8 | 47.1 |
> | Qwen3-8B + GPQA | 32.6 | 32.8 | 31.6 | 29.2 |
>
>
>
>
>
> - **[N/A] Iteration count $K$**: The iteration count $K$ is specified as $10$ on Line 314 of our manuscript. We did not conduct extensive ablations on this hyperparameter because our self-rewarding procedure follows prior work [1,2,3], as explicitly acknowledged on Line 236. In particular, reference [1] reports that $K = 10$ iterations are sufficient to yield stable and substantial self-improvement. Therefore, we believe this choice is well grounded in the existing literature, and no tuning was performed on our side.
>
> We are sincerely grateful for your comments, which further strengthened the rigor and completeness of our work! The additional experiments have now been incorporated into Appendix C.4 and C.5 of the updated manuscript PDF.

---

> ### Author Response · Authors · 2025-11-20
> **[Part 2/4] Author Response**
>
> > **`Weakness 3`**: The episodic buffer grows with experiences; managing retrieval, similarity search, etc might become onerous in production. More details on buffer capacity, retrieval latency, memory demands would strengthen the paper. Also, the retrieval process relies on semantic embedding. If the retrieval process is inaccurate, how does it influence the performance?
>
> Thank you very much for this careful and thoughtful question. We respectfully address each point in detail below:
>
> - **Buffer capacity**: As explicitly stated on Line 284 of the manuscript, after every batch of $T = 200$ test-time instances, the nighttime consolidation step is performed, after which the buffer is immediately cleared.
> - **Retrieval latency**: The retrieval in LatentEvolve is extremely fast: across a full daytime cycle, the average per-query retrieval latency is only **15.64 ms**, which is negligible relative to LLM forward costs.
> - **Memory demands**: Using Qwen2.5-7B as an illustrative example, the maximum memory required to store daytime experience triplets is approximately **0.56 GB**. This includes the context embedding $\mathbf{e}\_{c_j} \in \mathbb{R}^{3584}$, the base latent $\mathbf{z}\_{\text{base}} \in \mathbb{R}^{15 \times 3584}$, and the final latent $\mathbf{z}^* \in \mathbb{R}^{15 \times 3584}$. We believe this memory overhead is small, stable, and fully manageable in practical scenarios.
>
>
>
> Finally, regarding the concern about retrieval inaccuracies, we would like to clarify that LatentEvolve includes several mechanisms that jointly reduce the effect of imperfect retrieval:
>
> 1. The latent weaver transforms the base latent so that it moves closer to regions associated with successful reasoning, which naturally decreases sensitivity to noisy neighbors.
> 2. Retrieved latents are never merged directly; momentum transfer (Eq. (5)) incorporates only the optimization trajectory from past refinements rather than domain-specific content, providing a stable and broadly applicable signal.
> 3. The daytime self-rewarding step further adapts the latent representation to the current query, allowing residual retrieval noise to be corrected through task-driven refinement.
>
>
>
>
> We respectfully hope these clarifications help alleviate concerns regarding computational, memory constraints, and we sincerely appreciate the opportunity to elaborate.
>
>
>
>
> ---
> > **`Weakness 4`**: The self-rewarding strategy uses the model itself to score Q(y). There is an inherent risk of circularity or bias: if the model is wrong but confident, it may reinforce the wrong latent sequence. The paper would benefit from ablations or analysis of reward reliability.
>
>
> Thank you for this highly insightful question. We fully agree that relying on a model to score its own outputs raises concerns about circularity and reward bias. Fortunately, prior self-rewarding literature has already investigated this issue in depth and provides a compelling explanation, commonly referred to as the “lucky hit” phenomenon.
>
> **Theoretical grounding: the “lucky hit” effect**
>
> As formalized in TTRL [3], even if the model mispredicts an answer, the reward remains reliable as long as the model’s estimated label differs from its own predicted answer. In this case, the verifier naturally assigns a negative reward, which coincides with the correct supervision signal. In other words, the reward need not match the ground-truth label; it only needs to disagree with the model’s own incorrect prediction. **TTRL shows that this mechanism yields reward accuracy consistently above 80%** (reward accuracy indicates the proportion of self-rewards that match rewards computed from the ground-truth label), even when answer accuracy is much lower, demonstrating that self-rewarding remains surprisingly robust.
>
>
> **Empirical verification in LatentEvolve**
>
> To validate that LatentEvolve similarly benefits from the lucky-hit phenomenon, we conduct three reward-evaluation experiments:
>
>
> - Reward computed by the same model (self-reward).
> - Reward computed by a larger model (external verifier).
> - Reward computed by a larger model with access to ground truth (oracle).
>
> Below, we report how often the first two settings agree with the oracle rewards:
>
> *[Each cell determines the percentage of the given reward aligning with the oracle reward.]*
> |Setting|GSM8K|MATH|SciBench|JAMA|
> |-|-|-|-|-|
> |Qwen2.5-7B for rewarding|0.79|0.76|0.68|0.71|
> |Larger model (Qwen2.5-32B) for rewarding|0.83|0.87|0.80|0.82
> |Oracle|1|1|1|1|
>
> These results show that, even in challenging domains, LatentEvolve’s self-rewarding signals maintain a strong alignment with orcale rewards. This empirically confirms that LatentEvolve inherits the core robustness mechanism identified in prior work: the lucky-hit phenomenon allows the system to provide reliable gradients even when answer predictions may be imperfect.

---

> ### Author Response · Authors · 2025-11-20
> **[Part 3/4] Author Response**
>
> > **`Weakness 5`**: While some cross-domain results are given, one may ask how the system performs when the new queries are far from any prior buffer entries (i.e., the retrieval gives weak plus few relevant experiences). Does the system fallback gracefully or degrade badly? ...
>
> To address your concern, we would like to respond by the following two points:
>
> - **LatentEvolve already demonstrates strong cross-domain generalization.** As shown in Figure 3, when the system is evolved on MATH, it still yields a substantial improvement of 7.3 percent on JAMA, despite the two tasks belonging to **entirely different domains**: mathematical reasoning versus clinical reasoning. This indicates that the learned latent transformations are not narrowly domain-bound.
> - **We additionally evaluate LatentEvolve on an entirely new out-of-distribution benchmark**, i.e., FinancialReasoning [4], which assesses financial and business reasoning, and is completely OOD for MATH dataset. The results are summarized below.
>
> *[The experiments here follow the setting in Figure 3 (sequentially evolve LatentEvolve on MATH/MMLU). We add another OOD evaluation dataset, namely the FinancialReasoning.]*
> | | Test on MATH | Test on JAMA | Test on FinancialReasoning |
> |:-|:-:|:-:|:-:|
> | iter 1: MATH (200) | 65.0 | 35.2 | 61.3 |
> | iter 2: MATH (200) | 78.6 | 37.3 | 65.6 |
> | iter 3: MMLU (200) | 80.2| 39.0| 66.8 |
> | iter 4: MMLU (200) | 80.2 | 42.4 | 67.9 |
>
> We observe consistent gains across all three domains, including the newly introduced FinancialReasoning dataset. We humbly attribute this to nighttime consolidation learns latent-space transformations that transfer beyond the training domain, even when retrieved neighbors are not strongly aligned with the current task.
>
>
> ---
>
> > **`Question 1`**: Could you provide detailed statistics on the inference cost (wall-clock time, latency) of daytime scaling + retrieval + latent optimization vs. a vanilla inference pass? In deployment settings, latency is often critical.
>
> Thank you for this thoughtful comment! We hope to address your concern from two perspectives.
>
> - **The goal of Test-time Scaling is precisely to trade additional test-time computation for improved performance.** Prior TTS work [1,2] commonly incurs fivefold or even tenfold higher inference delays/FLOP budgets in pursuit of better accuracy. Before discussing concrete latency numbers, we wish to respectfully emphasize that a TTS method should not be expected to match the inference cost of a vanilla model. Rather, it should be evaluated by how effectively it converts a comparable computation budget into performance gains relative to other TTS approaches.
> - **Report the inference costs.** As shown below, under similar wall-clock budgets, LatentEvolve consistently yields substantially larger improvements than baselines. Compared with LatentSeek, LatentEvolve requires approximately **7%** more wall-clock time (6.2 h vs. 5.8 h), yet delivers a **17.2%** relative improvement in accuracy (77.60 vs. 66.20).
>
> *[Comparison conducted on Qwen2.5-7b model on a L40 (48GB) GPU.]*
> |Method|MATH||SciBench||
> |-|-|-|-|-|
> ||Wall-clock time|Perf.|Wall-clock time|Perf.|
> |Vanilla|2.5h|55.80|4.3h|11.27
> |LatentSeek|5.8h|66.20|8.1h|14.45
> |LatentEvolve|6.2h|77.60|9.6h|19.79|
>
>
> With due respect, we hope this analysis further clarifies LatentEvolve's efficiency within the test-time scaling landscape.
>
> ---
>
> > **`Question 2`**: If the system is exposed to a query from a domain completely unseen so far (so retrieval returns low‐similarity neighbours), how does performance degrade? Is there a fallback default initialization?
>
> We sincerely appreciate the reviewer’s thoughtful suggestions. For a detailed discussion, we respectfully refer the reviewer to our response to `Weakness 5`.

---

> ### Author Response · Authors · 2025-11-20
> **[Part 4/4] Author Response**
>
> > **`Question 3`**: Could you share some qualitative examples of latent sequences (zbase → z*), or visualise how the latent space evolves over time (e.g., clustering of queries, drift in latent interventions)? This would help understand what the system is learning.
>
>
> Thank you very much for this thoughtful question! We fully agree that understanding how the latent space evolves is important for interpreting what LatentEvolve learns. In response, we have added two analyses that directly visualize and quantify the latent-evolution process:
>
> **1. Visualization of latent evolution**
>
> Recall that in LatentEvolve, for each task query we first obtain $\mathbf{z}\_{\text{base}}$ via Eq. 4 (denoted *base latent*). The nighttime weaver then produces $\mathbf{z}'\_{\text{base}}$ (denoted *refined latent*), and after daytime optimization we obtain $\mathbf{z}^*$ (denoted *success latent* when the final trajectory is correct). We apply t-SNE to visualize the distributions of these three latent clusters on SciBench.
>
> The visualization results are placed in Appendix C.1. As shown, $\mathbf{z}\_{\text{base}}$ lies far from the success-latent region. Across Iterations 2–3, $\mathbf{z}'\_{\text{base}}$ progressively moves toward the success cluster, demonstrating that nighttime consolidation systematically pushes latent initializations toward regions associated with correct reasoning.
>
> **2. Quantitative drift of latent-cluster centroids.**
>
> To complement the visualization, we compute Euclidean distances between the centroids of the three clusters across daytime-nighttime iterations.
>
>
> *[Euclidean distance between latent-cluster centroids (on Llama-3.2-3B + JAMA). Iter 1–3 correspond to successive daytime–nighttime cycles.]*
> |Llama-3.2-3B + JAMA|Iter 1|Iter 2|Iter 3|
> |-|-|-|-|
> |base latent <-> success latent| 2317.401| 2466.180| 2289.177|
> |refined <-> success latent| - |1094.228| 834.102|
>
> *[Euclidean distance between latent-cluster centroids (on Qwen2.5-7B + MATH). Iter 1–2 correspond to successive daytime–nighttime cycles.]*
> |Qwen2.5-7B + MATH|Iter 1|Iter 2|
> |-|-|-|
> |base latent <-> success latent|1343.296|1308.065|
> |refined <-> success latent|-|955.109|
>
>
> As shown, across both settings, **the refined latents move steadily closer to the success-latent region** as daytime and nighttime phases alternate, whereas the base latents remain consistently farther away. We respectfully believe this quantitatively confirms that nighttime consolidation indeed supplies a significantly better initialization point for downstream reasoning.
>
> ---
>
>
> [1] Seek in the dark: Reasoning via test-time instance-level policy gradient in latent space
> [2] Self-rewarding language models
> [3] Ttrl: Test-time reinforcement learning
> [4] FinanceReasoning: Benchmarking Financial Numerical Reasoning More Credible, Comprehensive and Challenging
> [5] Scaling LLM Test-Time Compute Optimally can be More Effective than Scaling Model Parameters
> [6] s1: Simple test-time scaling

---

> ### Author Response · Authors · 2025-11-25
> **Thank you & Looking forward to further discussion**
>
> Dear Reviewer **8Ufs**,
>
> We deeply appreciate your dedication to engaging in author–reviewer discussions. Recognizing the existing discrepancies in the understanding of our manuscript, we have outlined your key concerns and our responses for enhanced communication:
>
> 1. **Regarding computational/runtime overhead `Weakness 1` `Question 1`**: We respectfully clarified the full daytime–nighttime computational cost across multiple model–dataset settings.
> 2. **Regarding hyperparameter sensitivity `Weakness 2`**: We provided both existing and newly added sensitivity studies, covering latent length, buffer size, retrieval metrics, and threshold τ.
> 3. **Regarding buffer scalability `Weakness 3`**: We numerically and empirically validate that the retrieval latency and memory footprint of episodic buffer remaining lightweight.
> 4. **Regarding reward reliability in self-rewarding `Weakness 4`**: We provided theoretical grounding and empirical evaluations demonstrating that self-rewarding remains reliable due to the “lucky-hit” mechanism proposed in TTRL.
> 5. **Regarding OOD generalization when retrieval is weak `Weakness 5` `Question 2`**: We demonstrated strong cross-domain transfer by including results on a completely OOD dataset FinancialReasoning benchmark.
> 6. **Regarding qualitative/visual analyses of latent evolution `Question 3`**: We added both t-SNE visualizations and centroid-drift measurements demonstrating that nighttime refinement systematically moves latent states toward regions associated with successful reasoning.
>
> For other issues not mentioned here, please refer to our detailed rebuttal response. We sincerely hope this addresses your concerns! We humbly look forward to further discussion with you.
>
> Warm regards,
> **Authors**

---

> > ### Comment · Reviewer_8Ufs · 2025-11-27
> > **Thank you for rebuttal**
> >
> > Thank you for your detailed responses. The rebuttal has addressed my concerns. I have upgraded my score to 6.

---

> > > ### Author Response · Authors · 2025-11-27
> > > **Thank you immensely!**
> > >
> > > We sincerely appreciate your thoughtful feedback and your supportive view of our work. We are delighted that our rebuttal successfully addressed your concerns!

---

### Official Review · Reviewer_hgWP · 2025-10-31

**Soundness:** 2
**Presentation:** 3
**Contribution:** 2
**Rating:** 2
**Confidence:** 4

**Summary:**

LatentEvolve is a novel self-evolving test-time scaling (TTS) framework for Large Language Models (LLMs) that aims to enable LLMs to progressively learn "how to scale test-time computation" without altering model parameters during inference. Inspired by the Complementary Learning System (CLS) theory of the human brain, LatentEvolve introduces a dual-phase evolutionary process comprising "Daytime Scaling" for rapid, episodic adaptation and "Nighttime Scaling" for slow, procedural consolidation. This allows LLMs to accumulate and refine experiential knowledge in an unsupervised manner, leading to enhanced reasoning capabilities.

While the mechanism of Test-Time Scaling using memory (latent representations) rather than computation (tokens/steps) is novel, the paper fails to contribute meaningful insight or theoretical understanding to the community. The Complex Learning System (CLS) analogy functions as an unnecessary, unsupported wrapper around a complex heuristic, and the authors fail to provide the necessary ablations or latent space analyses to demonstrate why the full, complex system is necessary.

**Strengths:**

1.  This paper is well-structured and easy to follow. The figures and tables are clear, professionally executed, and effectively convey the high-level architecture and quantitative results.
2.  LatentEvolve introduces a novel direction for Test-Time Scaling by evolving the memory (latent space representations) rather than scaling the computation (tokens/steps). This is a conceptually interesting shift from existing TTS methods like CoT or Self-Refine and is a valuable idea for the community.
3.  LatentEvolve achieves state-of-the-art results across several reasoning benchmarks, demonstrating its practical effectiveness.

**Weaknesses:**

1. The paper heavily relies on a biological metaphor (CLS) which does not translate into a strong, mathematically-justified framework. In other words, the connection between EvolveLatent and the hippocampus (Daytime Scaling) and neocortex (Nighttime Scaling)  is purely descriptive. If the approach truly brings insight, the paper should rigorously define what property of the latent space (e.g., lower variance, better separability, higher clustering of task-relevant information) is improved by the "Nighttime" consolidation, independent of the bio-analogy. Without this, the CLS is an unnecessary wrapper around a complex heuristic.
2. The method combines multiple components: A latent memory buffer, a retrieval mechanism, and an integration/consolidation step. An ablation study is needed where "Nighttime Scaling" is replaced by a simple, computationally cheap Exponential Moving Average (EMA) or FIFO (First-In, First-Out) replacement strategy. If the performance gap is marginal, the core mechanism of "evolution" is debunked.
3. The method's performance will inevitably depend on the size of the latent memory buffer. I wonder if the memory needs to grow indefinitely to maintain performance gains across a wide distribution of tasks (e.g., across hundreds of thousands of prompts). If so, the method is not scalable for real-world deployment.
4. The experimental section lacks depth regarding the _mechanism_ of the proposed method's success, failing to generate insight. The core novelty resides in "self-evolving test-time scaling in latent space," yet there is no visual evidence of this evolution. Does Nighttime truly consolidate the latent space, making representations for similar problems tighter or better clustered? Does the retrieval process pull the currently processed latent vector closer to the centroid of successful historical solutions? Without these analysis, the claims of "evolution" and "consolidation" remain hand-wavy and unproven.

**Questions:**

- How is the performance degradats if the memory buffer size is drastically limited (e.g., using only the 100 or 500 most recent examples)?

---

> ### Author Response · Authors · 2025-11-20
> **[Part 1/3] Author Response**
>
> We sincerely thank you for the thoughtful and constructive reviews of our manuscript! Based on your questions and recommendations, we give point-by-point responses to your comments and describe the revisions we made to address them.
>
>
> ---
>
> > **`Weakness 1`**: If the approach truly brings insight, the paper should rigorously define what property of the latent space (e.g., lower variance, better separability, higher clustering of task-relevant information) is improved by the "Nighttime" consolidation, independent of the bio-analogy. Without this, the CLS is an unnecessary wrapper around a complex heuristic.
>
>
>
> Thank you for this crucial question, which genuinely helps us further highlight the contributions of the paper. To address your concern clearly, we begin by restating what nighttime consolidation provides:
>
> (i) a reasoning initialization that lies closer to the success manifold, and
> (ii) richer, task-relevant latent information.
>
> We then substantiate these properties through three groups of experiments.
>
> **Exp 1: Visualization of latent-reasoning evolution**
>
> Recall that in LatentEvolve, for each task query we first obtain $\mathbf{z}\_{\text{base}}$ via Eq. 4 (denoted **base latent**). The nighttime weaver then produces $\mathbf{z}'\_{\text{base}}$ (denoted **refined latent**), and after daytime optimization we obtain $\mathbf{z}^*$ (denoted **success latent** when the final trajectory is correct). We apply t-SNE to visualize the distributions of these three latent clusters on JAMA dataset. The full visualization can be found in Appendix C.1 of the updated manuscript.
>
> As shown, the base latents remain far from the success-latent cluster. Across iterations 2 and 3, the refined latents produced by the nighttime weaver move progressively closer to the success-latent region. This clearly indicates that, **through repeated nighttime scaling, the weaver learns to supply the model with reasoning initializations that lie nearer to regions associated with successful inference.**
>
>
> **Exp 2: Quantifying distances between latent clusters**
>
> While Exp 1 visually demonstrates that nighttime consolidation gradually steers the latent representations toward the success manifold, it does not yet provide numerical evidence. Therefore, we compute the Euclidean distances between the centroids of the three latent clusters at different iterations. The results are summarized in the tables below and also updated in Appendix C.2:
>
>
> *[Euclidean distance between latent-cluster centroids (on Llama-3.2-3B + JAMA). Iter 1–3 correspond to successive daytime–nighttime cycles.]*
> |Llama-3.2-3B + JAMA|Iter 1|Iter 2|Iter 3|
> |-|-|-|-|
> |base latent <-> success latent| 2317.401| 2466.180| 2289.177|
> |refined <-> success latent| N/A  (weaver not trained yet)|1094.228| 834.102|
>
> *[Euclidean distance between latent-cluster centroids (on Qwen2.5-7B + MATH). Iter 1–2 correspond to successive daytime–nighttime cycles.]*
> |Qwen2.5-7B + MATH|Iter 1|Iter 2|
> |-|-|-|
> |base latent <-> success latent|1343.296|1308.065|
> |refined <-> success latent|N/A|955.109|
>
>
> As shown, across both settings, **the refined latents move steadily closer to the success-latent region** as daytime and nighttime phases alternate, whereas the base latents remain consistently farther away. We respectfully believe this quantitatively confirms that nighttime consolidation indeed supplies a significantly better initialization point for downstream reasoning.
>
>
> **Exp 3: Logistic-regression probe on task-relevant information**
>
> To assess whether nighttime consolidation enriches task-relevant information, we conduct a linear-probe study at iteration 2. For each query, we extract two representations: (1) the base latent, and (2) the refined latent produced by the nighttime weaver.
>
> We then train **two separate** lightweight classifiers, each using a single MLP-based logistic head: one on the vanilla base latents and one on the refined latents. Each classifier predicts whether the final reasoning trajectory succeeds or fails. Because the probe is deliberately minimal, differences in accuracy directly reflect how much linearly decodable task information each latent type contains.
>
> As shonw below, refined latents achieve consistently higher prediction accuracy, indicating that nighttime consolidation indeed injects more task-salient structure that even simple linear models can reliably decode.
>
>
> |Dataset|Qwen2.5-7b+MATH|Gemma3 + SciBench|Llama3.2-3B+JAMA|
> |-|-|-|-|
> |Probe acc (from base latent)|73.8|12.6|40.1|
> |Probe acc (from refined latent)|76.2|20.3|48.6|
>
>
>
> We respectfully believe this provides complementary evidence that nighttime consolidation enriches the latent manifold with meaningful problem-specific signals.

---

> ### Author Response · Authors · 2025-11-20
> **[Part 2/3] Author Response**
>
> > **`Weakness 2`**:  An ablation study is needed where "Nighttime Scaling" is replaced by a simple, computationally cheap Exponential Moving Average (EMA) or FIFO (First-In, First-Out) replacement strategy. If the performance gap is marginal, the core mechanism of "evolution" is debunked.
>
> To address your concern, we would like to (1) respectfully clarify that our manuscript already includes an ablation isolating the contribution of nighttime scaling, and (2) additionally provide the FIFO baseline you suggested.
>
> **1. Nighttime scaling has already been ablated in Figure 4**
>
> As shown in Figure 4, when we remove nighttime scaling and simply allow the episodic buffer to grow indefinitely (noting that, in the original LatentEvolve, the buffer is cleared after every nighttime consolidation), performance drops substantially: SciBench decreases by $6.8\%$ and MATH by $12.3\%$. This demonstrates that merely enlarging the buffer without performing deep consolidation is insufficient.
>
>
> **2. We additionally include the FIFO-buffer baseline you proposed**
>
> In the original “w/o nighttime” baseline, the buffer is allowed to accumulate indefinitely. Following your suggestion, we further implement a FIFO baseline that keeps the episodic buffer capped at $T=200$ trajectories. The results are reported below:
>
> *[We follow the setting the Figure 4, with $L'$ set as 30. "w/ nighttime" denotes disgarding the nighttime consodliation and allows the buffer to grow infinitely; "w/ nighttime + FIFO daytime" denotes the buffer is fixed with a 200 budget and managed by FIFO.]*
> |Setting|MATH|SciBench|
> |-|-|-|
> |Full LatentEvolve|80.3|33.4|
> |w/ nighttime|68.4|26.6|
> |w/ nighttime + FIFO daytime|66.1|24.0|
>
> We observe that the FIFO baseline performs not only far worse than LatentEvolve, but even worse than the w/o nighttime baseline. This is intuitive: because FIFO keeps only a small sliding window of experiences and lacks nighttime consolidation, the system is unable to accumulate and internalize the long-range latent structure required for improvement.
>
> We humbly hope these results clarify why nighttime scaling is essential and why simple buffer-management strategies cannot replace it.
>
> ---
>
> > **`Weakness 3`**: The method's performance will inevitably depend on the size of the latent memory buffer. I wonder if the memory needs to grow indefinitely to maintain performance gains across a wide distribution of tasks (e.g., across hundreds of thousands of prompts). If so, the method is not scalable for real-world deployment.
>
>
> We would like to respectfully clarify a potential misunderstanding: **the episodic buffer in LatentEvolve does not grow indefinitely**. As stated on Line 292, the buffer is *refreshed* after every nighttime scaling phase, in other words, it is cleared entirely.
>
> For this reason, LatentEvolve does **not** "needs to grow indefinitely to maintain performance gains across hundreds of thousands of prompts". (In fact, as shown in our ablation study in Section 5.4, removing the nighttime phase and retaining only a continually growing memory buffer leads to worse performance.) Instead, the system continuously distills experience into the weaver, keeping the buffer compact and efficient at all times.
>
>
> To prevent any potential confusion, we have added clarifying text at Line 292. Thank you again for raising this critical point!

---

> ### Author Response · Authors · 2025-11-20
> **[Part 3/3] Author Response**
>
> > **`Weakness 4`**: The experimental section lacks depth regarding the mechanism of the proposed method's success, failing to generate insight. The core novelty resides in "self-evolving test-time scaling in latent space," yet there is no visual evidence of this evolution. Does Nighttime truly consolidate the latent space, making representations for similar problems tighter or better clustered? Does the retrieval process pull the currently processed latent vector closer to the centroid of successful historical solutions? Without these analyses, the claims of "evolution" and "consolidation" remain hand-wavy and unproven.
>
> Thank you for this thoughtful question, which helps make our contributions clearer and more intuitive. In our response to `Weakness 1`, we:
>
> - **visually** show in Figure 5 that latent representations produced after nighttime scaling progressively move closer to the successful reasoning region;
> - **quantitatively** demonstrate in Appendix C.2 that nighttime scaling enables the model to begin reasoning from states increasingly aligned with successful outcomes;
> - **quantitatively** show that latent vectors generated by the nighttime weaver encode substantially more task-relevant information.
>
> In addition, Figure 5 also shows that momentum transfer also pulls the weaver-generated latents closer to the success region, illustrating that momentum-based experience transfer is reasonable and effective.
>
>
> Taken together, we believe these results provide strong evidence for the “evolution” and “consolidation” properties central to LatentEvolve.
>
>
> ---
>
> > **`Question 1`**: How is the performance degraded if the memory buffer size is drastically limited (e.g., using only the 100 or 500 most recent examples)?
>
>
> In our response to `Weakness 3`, we clarified that the episodic buffer in LatentEvolve is inherently **limited** (in practice, $T = 200$). After every $T = 200$ trajectories, nighttime consolidation is performed and the buffer is subsequently cleared.
>
> In addition, we would like to point out that the original manuscript already includes a sensitivity study on the buffer size $T$ in Appendix B.6 (Table 6), covering $T \in \\{200, 300, 500, 1000\\}$. In summary, smaller $T$ values enable more frequent iterative refinements within the same training budget, yielding smoother convergence, whereas larger $T$ values reduce the number of refinement cycles, thereby coarsening the evolutionary process. For a more detailed analysis, we respectfully refer the reviewer to Appendix B.6.

---

> ### Author Response · Authors · 2025-11-25
> **Thank you & Looking forward to further discussion**
>
> Dear Reviewer **hgWP**,
>
> We sincerely appreciate your thoughtful and constructive insights. To facilitate clearer communication, we summarize your key concerns and our corresponding clarifications below:
>
> 1. **Regarding latent-space insight `Weakness 1` `Weakness 4`**: We clarified—through visualization, distance quantification, and linear-probe analyses—that nighttime consolidation moves latent states closer to the success manifold and enriches task-relevant structure.
> 2. **Regarding simple baselines vs. evolution `Weakness 2`**: We provided both existing and newly added ablations (including a FIFO buffer) showing that nighttime evolution is indeed irreplaceable.
> 3. **Regarding memory scalability `Weakness 3` `Question 1`**: We clarified that the episodic buffer never grows indefinitely, as it is fully cleared after each nighttime phase.
>
> For other issues not mentioned here, please refer to our detailed rebuttal response. We sincerely hope this addresses your concerns! We humbly look forward to further discussion with you.
>
> Warm regards,
> **Authors**

---

### Official Review · Reviewer_iXSM · 2025-11-01

**Soundness:** 3
**Presentation:** 4
**Contribution:** 3
**Rating:** 8
**Confidence:** 3

**Summary:**

This paper addresses a key limitation of existing TTS methods: they are static, independent processes that do not improve or adapt over time. The authors reframe TTS as a skill that an LLM can learn and master. They propose LatentEvolve, a novel self-evolving framework inspired by the CLS theory from neuroscience. The method operates on a "Daytime-Nighttime" cycle, where during the daytime (inference), the model performs inference by fusing two parallel latent scaling modules, and nighttime (evolution) is an offline phase, which does self-distillation. Through this cycle, the stable module progressively "evolves," allowing the model to get better at the skill of scaling its computation. Experiments on reasoning benchmarks (GSM8K, MATH, SciBench) show that LatentEvolve significantly outperforms both base models and existing static TTS methods like Self-Refine and Time-Warping.

**Strengths:**

- The primary strength is the novel reframing of Test-Time Scaling as a learnable, evolving skill rather than a static inference algorithm. This is a significant conceptual leap.
- Instead of an ad-hoc design, this methodology is principled in CLS.
- The empirical results are quite strong.

**Weaknesses:**

- The paper clearly demonstrates the benefits of the Nighttime evolution steps but does not quantify their cost. For practical application, it's crucial to know the computational and data requirements of a single offline evolution phase. Could the authors please quantify the computational cost (e.g., GPU hours, number of samples required) for a single "Nighttime" evolution step?
- The paper states the transient module is "reborn," which implies re-initialization. It is not fully clear if this is a full random re-initialization (which might be unstable) or a reset to a pre-trained state. Could you please clarify the "reborn" process for the Transient Latent Scaling (TLS) module? Is it a full re-initialization with random weights, or is it reset to a specific pre-trained checkpoint?

**Questions:**

See weaknesses.

---

> ### Author Response · Authors · 2025-11-20
> **[Part 1/1] Author Response**
>
> Thank you immensely for your time and efforts, as well as the helpful and constructive feedback! Here, we give point-by-point responses to your comments.
>
>
> ---
>
> > **`Weakness 1`**: The paper clearly demonstrates the benefits of the Nighttime evolution steps but does not quantify their cost. For practical application, it's crucial to know the computational and data requirements of a single offline evolution phase. Could the authors please quantify the computational cost (e.g., GPU hours, number of samples required) for a single "Nighttime" evolution step?
>
> Thank you for this important question! We respectfully provide a detailed breakdown of the nighttime computational cost across different LLM–dataset configurations as follows.
>
> *[Llama-3.2-3B + JAMA. “Iter 1” denotes one full cycle of daytime + nighttime scaling, each iteration using $T=200$ test samples (explicitly stated on Line 284).]*
> | Metrics | Iter 1 | Iter 2 | Iter 3 |
> |---------|--------|--------|--------|
> | Nighttime GPU hours (h) | 1.23h | 1.11h | 1.33h |
> | Daytime wall-clock time | 2.5h | 2.5h | 2.4h |
> | Nighttime training samples | 127 | 133 |  139|
> | Performance (Acc.) | 46.8% | 47.2% | 47.9% |
>
> *[Qwen2.5-7B + MATH. Each iteration again uses $T=250$ test samples.]*
> | Metrics | Iter 1 | Iter 2 |
> |---------|--------|--------|
> | Nighttime GPU hours (h) | 1.3h | 1.3h |
> | Daytime wall-clock time | 1.8h | 1.8h |
> | Nighttime training samples | 188 | 213 |
> | Performance (Acc.) | 74.2% |77.8%  |
>
> As shown, the nighttime resource requirements for LatentEvolve remain moderate across configurations, while the framework provides substantial performance improvements in a fully unsupervised manner. We sincerely hope this addresses your concerns regarding nighttime computational efficiency.
>
>
>
> ---
> > **`Weakness 2`**: The paper states the transient module is "reborn," which implies re-initialization. It is not fully clear if this is a full random re-initialization (which might be unstable) or a reset to a pre-trained state. Could you please clarify the "reborn" process for the Transient Latent Scaling (TLS) module? Is it a full re-initialization with random weights, or is it reset to a specific pre-trained checkpoint?
>
>
> Thank you for your thoughtful question. We understand that your concern may relate to the “refresh” operation described on Line 292. In practice, this operation simply clears the episodic buffer after nighttime consolidation and does not modify model parameters, nor does it introduce any instability.
>
>
> To make this transition fully transparent, we respectfully re-clarify how the different components of LatentEvolve behave across the daytime and nighttime phases:
>
> | Phase  | Episodic Buffer $\mathcal{M}^{(t)}$| Latent Weaver $\mathbf{W}_\psi$  |
> | - | - | - |
> | **Start of the $t$-th daytime scaling**| Buffer is **empty**  | Uses parameters updated from previous nighttime; integrates information from $\mathcal{M}^{(t-1)}$; parameter count unchanged |
> | **End of the $t$-th daytime / Start of $t$-th nighttime** | Buffer has accumulated adaptation triplets $\{(\mathbf{e}_{c_j}, \mathbf{z}_\text{base}, \mathbf{z}_j^*)\}_{idx=1}^{\|\mathcal{M}^{(t)}\|}$ with size < 200 | **Unchanged** (no update yet) |
> | **End of the $t$-th nighttime scaling**| Buffer is **cleared/reset** after training with Eq. (7)                                                        | Weaver is **updated** using the collected adaptation triplets according to Eq. (7) |
> | **Next iteration**  | New empty buffer for iteration $t{+}1$ | .. |
>
> Accordingly, we respectfully argue that all components are updated in a **stable and consistent** manner. The only re-initialization occurs when the episodic buffer is cleared after each nighttime phase, which does not introduce instability; its sole purpose is to prevent unbounded memory growth. We sincerely hope this resolves your concern.

---

> ### Author Response · Authors · 2025-11-25
> **Thank you & Looking forward to further discussion**
>
> Dear Reviewer **iXSM**,
>
> We deeply appreciate your dedication to engaging in author–reviewer discussions. Recognizing the existing discrepancies in the understanding of our manuscript, we have outlined your key concerns and our responses for enhanced communication:
>
> 1. **Regarding nighttime computational cost `Weakness 1`**: We clarify the full GPU-hour and sample requirements for each nighttime evolution step across multiple configurations.
> 2. **Regarding the “reborn” operation `Weakness 2`**: We respectfully clarify that “reborn” refers only to clearing the episodic buffer after nighttime consolidation, while all model parameters remain stable and are never re-initialized.
>
> For other issues not mentioned here, please refer to our detailed rebuttal response. We sincerely hope this addresses your concerns! We humbly look forward to further discussion with you.
>
> Warm regards,
> **Authors**

---

> > ### Comment · Reviewer_iXSM · 2025-11-27
> > **Response by Reviewer iXSM**
> >
> > All my queries have been resolved and I would like to maintain my positive score (8).

---

> > > ### Author Response · Authors · 2025-11-27
> > > **Thank you immensely!**
> > >
> > > Thank you for your kind comment and strong support of our work. We deeply appreciate your reviews and feedback on the computational cost and the refresh operation of LatentEvolve, which have elevated the quality of our work!

---

### Comment · Area_Chair_Hb5p · 2025-11-24
**Discussions are encouraged**

Dear Reviewers,

The authors have submitted their rebuttal addressing your concerns. Would you please kindly check the authors' responses and check whether your core concerns are adequately addressed, and adjust your initial score/comment if necessary? Your feedback is critical for the final decision. Thank you for your rigorous review and time!

Best regards,
Your AC

---

### Public Comment · ~Ssomeone1 · 2025-11-27
**Questions to authors about baseline scores**

Dear Authors,

Thank you for your interesting work on test-time scaling. I have some questions regarding the experimental results reported in your paper that I hope you could kindly clarify:

1. Baseline performance discrepancies: I noticed **substantial differences** between the baseline scores reported in **Tables 1 and 2** and those in the **official technical reports**. For instance:

    * On **MATH-500**: Your reported Qwen2.5-7B score is 55.80 versus 77.6 in the official report; Qwen3-8B is 55.20 (73.80 with **LatentEvolve**) versus 87.4 (non-thinking) and 97.4 (thinking) officially.
    * On **AIME 24**: Your reported scores are 0 for Qwen2.5-7B and 3.33 for Qwen3-8B, improving to 23.33 and 26.67 with **LatentEvolve**, whereas official scores are 9.1 and 29.1 (non-thinking) / 76.0 (thinking) respectively.
    * on **GPQA**: Your reported scores are 34.85 for Qwen3-4B and 28.82 for Qwen3-8B, improving to 38.89 and 32.82 with **LatentEvolve**, whereas official scores are 41.7 (non-thinking) / 55.9 (thinking) and 39.3 (non-thinking) / 62.0 (thinking) respectively.
    * ...
*While I notice you added AIME Avg@32 results for Qwen2.5-7B and Llama3.2-3B in the revised version, these scores still differ drastically from those in the main experimental tables (where the 0-score baselines appear misleading), and the multi-run results for other models remain unclear.*

2. Scoring patterns on AIME: I observed that all reported **AIME 24/25** scores appear to be multiples of **1/30 (3.33%)**. Given that AIME contains 30 questions and the recommended evaluation protocol involves averaging over 10 runs, could you clarify whether your results represent single-run evaluations? Alternatively, did all models coincidentally **achieve identical scores** across all 10 runs in your experiments, such that the averaged results still yield exact multiples of 1/30 (e.g., x.33%, x.67%)?

3. Do you employ some **sophisticated evaluation protocol** that systematically deflates all baseline scores? Alternatively, are you suggesting that the evaluation methodologies used by Qwen and other LLM teams contain **fundamental errors**?

I appreciate your time in addressing these questions and look forward to your clarification.

**References:**
* Qwen3 Technical Report (https://arxiv.org/abs/2505.09388)
* https://artificialanalysis.ai/methodology/intelligence-benchmarking

---

### Meta-Review · Area_Chair_xmFs · 2025-12-22

**Summary:**

This paper proposes LatentEvolve, a self-evolving test-time scaling framework that treats TTS as a skill an LLM can gradually learn rather than a static, per-query process. The method alternates between a daytime inference phase using latent-space retrieval and optimization and a nighttime offline self-distillation phase that consolidates past inference experience, allowing performance to improve over time without changing model parameters at inference.

**Reviewer Concerns:**

The following concerns are still outstanding:

(1) The experimental section lacks depth regarding the mechanism of the proposed method's success, failing to generate insight. (Although the author states the results, the author did not explain how or why this movement produces better reasoning.)

(2) Several concerns related to the retrieval process remain unaddressed. For example, it is unclear whether the retrieval mechanism actually pulls the currently processed latent vector closer to the centroid (or core region) of successful historical solutions. In addition, the paper does not analyze how inaccuracies in retrieval affect performance. I did not find specific answers to these questions in the rebuttal, which makes it difficult to assess the role of the retrieval component.

**Reviewer Scores:**

Reviewer iXSM (score: 8) provided brief comments and is likely to keep their score.

Reviewer hgWP (score: 2) is mainly concerned about the lack of rigorous analysis and clear intuition for the method, as well as the absence of general insights, and is therefore unlikely to increase their score.

Reviewer 8Ufs (score: 4) indicated an increase to 6.

Reviewer HDXy (score: 6) indicated that they are likely to keep their score.

---

### Decision · Program_Chairs · 2026-01-26

Reject